

# Seasonal Patterns in Phytoplankton Biomass across the Northern and Deep Gulf of Mexico: A Numerical Model Study

Fabian A. Gomez[1,2], Sang-Ki Lee[2], Yanyun Liu[3,4], Frank J. Hernandez Jr.[1], Frank E. Muller-Karger[5], and John T. Lamkin[6]

[1] Division of Coastal Sciences, University of Southern Mississippi, Ocean Springs, MS, USA
[2] Atlantic Oceanographic and Meteorological Laboratory, NOAA, Miami, FL, USA
[3] Climate Prediction Center, NOAA/NWS/NCEP, College Park, MD, USA
[4] Innovim, LLC, Greenbelt, MD, USA
[5] College of Marine Science, University of South Florida, St Petersburg, FL, USA
[6] Southeast Fisheries Science Center, NOAA, Miami, FL, USA

*Correspondence to*: Fabian A. Gomez (fabian.gomez@noaa.com)

**Abstract.** Biogeochemical models that simulate realistic lower trophic levels dynamics, including the representation of main phytoplankton and zooplankton functional groups, are valuable tools for our understanding of natural and anthropogenic disturbances in marine ecosystems. However, previous three-dimensional biogeochemical modeling studies in the northern and deep Gulf of Mexico (GoM) have used only one phytoplankton and one zooplankton type. To advance our modeling capability of the GoM ecosystem and to investigate the dominant spatial and seasonal patterns phytoplankton biomass, we configured a 14-component biogeochemical model that explicitly represents nanophytoplankton, diatoms, micro-, and mesozooplankton. Our model outputs compare well with satellite and in situ observations, reproducing dominant seasonal patterns in chlorophyll and primary production. The model results show that diatom growth is strongly silica limited (>95%) in the deep GoM, and both nitrogen and silica limited (30-70%) in the northern shelf. Nanophytoplankton growth is weakly nutrient limited in the Mississippi delta year-round (<20%), and strongly nutrient limited in the deep GoM during summer (~80%). Such nutrient limitation patterns influence the spatial and seasonal phytoplankton composition, with the mean diatom to total phytoplankton biomass ratio ranging from ~0.5 near the Mississippi delta to <0.1 in the deep GoM. The examination of primary production and net phytoplankton growth indicates that the biomass losses, mainly due to zooplankton grazing, play an important role modulating the seasonal biomass patterns of the nanophytoplankton and diatoms. Our analysis further shows that the dominant physical process influencing the net phytoplankton growth is horizontal advection in the northern shelf, and vertical diffusion in the deep GoM. This study highlights the importance of representing small and large size plankton dynamics to describe primary production patterns, and emphases the needs for an integrated analysis of biologically and physically driven biomass fluxes to better understand phytoplankton biomass phenologies in the GoM.



# 1 Introduction

The Gulf of Mexico (GoM) is characterized by large spatial differences in plankton productivity and biomass, ranging from the oligotrophic Loop Current to the highly productive northern shelf. Productivity in this last region is strongly influenced by river run-off. The Mississippi-Atchafalaya (MS-A) River System is the largest river input with a mean river discharge of 21,524 $m^3$ $s^{-1}$ (Aulenbach et al., 2007), contributing more than 80% of the entire dissolved inorganic nitrogen (DIN) load into the northern GoM (Xue et al., 2013). The large plankton production and vertical stratification driven by the MS-A river system discharge promotes the development of a hypoxic bottom layer a few meters thick off Louisiana and Texas during summer (Obenour et al., 2013). This hypoxic layer, or "dead zone", is a source of major scientific and societal concerns, because of its deleterious impact on the coastal ecosystem. The influence of river runoff on plankton production substantially decreases offshore (Xue et al., 2013). In the oligotrophic deep GoM, the spatiotemporal patterns in phytoplankton biomass are mainly associated with seasonal changes in thermal stratification and mesoscale ocean dynamics (e.g. Muller-Karger et al., 2015).

Multiple ocean-biogeochemical modeling studies have been conducted in the northern GoM to understand the drivers of phytoplankton biomass variability, carbon export, nutrient cycling, and bottom hypoxia variability. Green et al. (2008) configured a zero-dimensional Lagrangian model of the Mississippi (MS) river plume, which included two types of phytoplankton (small and large size), two types of zooplankton (micro- and mesozooplankton), bacteria, detritus, ammonium and nitrate. This study derived distinct production patterns for small and large size phytoplankton production, concluding that primary production was mainly limited by physical dilution of nitrate, light attenuation, and the sinking of diatoms (large phytoplankton). More complex modeling efforts for the region include a series of three-dimensional (3-D), fully coupled ocean-biogeochemical models, based on Fennel's biogeochemical model (Fennel et al., 2006). The original Fennel's model formulation included ammonium, nitrate, phytoplankton, zooplankton (representing mesozooplankton), and two detritus types as state variables. Fennel et al. (2011) examined the underlying factors determining seasonal patterns in phytoplankton biomass in the Louisiana-Texas shelf, and concluded that phytoplankton production was not nitrogen limited near the MS delta. They also showed that zooplankton grazing played an important role in defining phytoplankton biomass changes, and speculated that physical transport of phytoplankton could impact biomass seasonality. Xue et al. (2013) configured Fennel's model for the entire GoM, describing main spatiotemporal patterns in plankton biomass and DIN in the coastal and oceanic domains. However, since they did not investigate underlying drivers (production, biomass losses) of phytoplankton biomass as was done in Fennel et al. (2011), less is know about the factors modulating the seasonality of phytoplankton in the deep GoM.

Significant differences in plankton production and carbon export can be expected between food webs dominated by small-size (nanophytoplankton, microzooplankton) and large-size (diatoms, mesozooplankton) plankton components. Sedimentation rates are enhanced (decreased) in diatom (small phytoplankton) based food webs, and therefore changes in phytoplankton composition could influence bottom remineralization processes (Dortch and Whiteledge, 1992; Dagg et al.,



2003; Green et al., 2008; Zhao and Quigg, 2014). In addition, changes in phytoplankton composition may modulate trophodynamics, which can impact the reproductive success of upper trophic levels, and therefore modulate marine population abundance (Rykaczewski and Checkley, 2008). In the GoM, 3-D regional ocean-biogeochemical models that include more than one plankton functional group have been implemented only for the western Florida shelf (Walsh et al., 2003). New modeling efforts adding complexity to the representation of lower-trophic level dynamics are required for the northern and deep GoM. A key modeling aspect is the characterization of diatoms and nanophytoplankton growth. It is well known that: 1) nanophytoplankton uptake nutrients more efficiently than diatoms; 2) diatoms can achieve greater growth rates than nanophytoplankton in nutrient-rich environments; and 3) diatoms require silicate as an additional nutrient for frustule formation (Kishi et al., 2007; Falkowski and Oliver, 2007). These differences should be considered when simulating phytoplankton responses to changes in nutrient availability.

The present study examines underlying factors determining spatial and seasonal patterns in phytoplankton biomass across the coastal and ocean domains in the GoM, using an ocean-biogeochemical model that explicitly simulates small- and large-size plankton groups. After validating the model results with available observations, we examine main seasonal patterns of phytoplankton biomass. Our main goals are to: 1) to describe the spatiotemporal patterns in growth limitation for diatoms and nanophytoplankton; and 2) to evaluate the coupled role of biological (phytoplankton production and biological losses) and physical (advection and diffusion of biomass) processes as drivers of phytoplankton seasonality. This study complements Fennel et al. (2011) on phytoplankton variability in the northern GoM, by adding complexity to the modeled lower-trophic level dynamics, extending the description of phytoplankton growth-limitation patterns to the deep GoM, and quantifying the role of advection and diffusion.

## 2 Data and model

### 2.1 Data

Monthly mean composite fields of SeaWiFS chlorophyll (1998-2011), and MODIS SST and chlorophyll-a (2003-2014), were retrieved from the Institute for Marine and Remote Sensing, University of South Florida (http://imars.usf.edu). These data were processed using standard NASA algorithms. SeaWiFS and MODIS Aqua satellite data were obtained either through direct broadcast from the satellite and captured with a ground antenna (e.g., SeaWiFS data), or from the NASA Goddard Space Flight Center. All products followed the latest implementation of the atmospheric correction based on Ding and Gordon (1995). Chlorophyll-a concentration from CZCS, SeaWiFS and MODIS was estimated using the NASA OC4 and OC3 band ratio algorithms (O'Reilly et al., 2000).

### 2.2 Model description

The biogeochemical model simulates nitrogen (N), silica (Si), and oxygen ($O_2$) cycling. The model includes 14-components: nitrate ($NO_3$), ammonium ($NH_4$), nanophytoplankton (small phytoplankton, PS), diatom (large phytoplankton,




PL), chlorophyll of nanophytoplankton and diatom (ChlS and ChlL), microzooplankton (small zooplankton, ZS), mesozooplankton (large zooplankton, ZL), small and large detritus (DS and DL), opal, labile dissolved organic nitrogen (DON), silicate (SiOH$_4$), and O$_2$. Small detritus is particulate nitrogen linked to SZ egestion and small plankton (PS + ZS) mortality, while large detritus is particulate nitrogen associated with LZ egestion and large plankton (PL + ZL) mortality.

Opal is non-living particulate Si linked to diatom mortality and zooplankton egestion. The state variables NO$_3$, NH$_4$, PS, PL, ZS, ZL, DS, DL, and DON are simulated in terms of mmol N m$^{-3}$, silicate and opal in terms of mmol Si m$^{-3}$, ChlS and ChlL in terms mg chlorophyll m$^{-3}$, and oxygen in mmol O$_2$ m$^{-3}$. The model does not include phosphate as limiting nutrient for phytoplankton growth. Although previous modeling studies suggested moderate phosphate limitation near the MS-A deltas, mostly limited to May-July (Laurent et al., 2012), we focus here on the role of N and Si, as observational studies strongly

suggest that N and Si are critical in modulating phytoplankton production and composition across the northern GoM (Dortch and Whitledge, 1992; Nelson and Dortch, 1996; Lohrenz et al., 1997; 2008; Rabalais et al., 2002; Zhao and Quigg, 2014).

The model describes the following processes: 1) phytoplankton growth as a function of temperature, light, NO$_3$ and NH$_4$, including NH$_4$ inhibition of NO$_3$ uptake; 2) silicate limitation of PL growth; 3) photo-acclimation, 4) phytoplankton exudation, 5) ZS grazing on PS and PL, 6) ZL grazing on PS and PL, and predation on ZS, 7) zooplankton egestion and

zooplankton excretion, 8) phytoplankton and zooplankton mortality, 9) nitrification, 10) detritus, DON and opal remineralization, 11) detritus, diatoms, and opal sinking, 12) sediment coupled nitrification/denitrification (instantaneous remineralization), and 13) oxygen production and consumption. Processes 1, 3, 9, and 12-13 follow Fennel *et al.* (2006; 2013) formulations, while processes 2, 4-8, and 10-11 follow Kishi *et al.* (2007) formulations. Descriptions of the model equations and parameters are included in Appendix A. Model parameter values are presented in Table 1.

The model domain encompasses the entire GoM and is based on the Regional Ocean Model System (ROMS) (Shchepetkin and McWilliams, 2005). The model's horizontal resolution is about 8 km and has 37 sigma-coordinate (bathymetry-following) vertical levels. Boundary conditions are Flather (Flather, 1976) and Chapman (Chapman, 1985) for the barotropic velocity and free surface, respectively, and a combination of radiation and nudging for the baroclinic velocity and tracers (Marchesiello et al., 2001). The open boundary nudging timescale is 4 days for the incoming signal and 90 days

for the outgoing signal. A third order upstream scheme and a fourth order Akima scheme are used for horizontal and vertical momentum advection, respectively. Multidimensional positive definitive advection transport algorithm (MPDATA) is used for horizontal and vertical tracer advection (Smolarkiewicz and Margolini, 1998). Horizontal viscosity and diffusivity are set to 1 m$^2$ s$^{-1}$, increasing gradually to 4 m$^2$ s$^{-1}$ in a 100 km wide sponge layer at the open boundaries to reduce signal reflection problems. Mellor and Yamada 2.5-level closure scheme is used for vertical turbulence (Galperin *et al.*, 1988). Initial and

open boundary conditions are derived from a 25 km-resolution basin-scale model (Liu et al., 2015). Surface heat flux and wind stress are estimated using bulk parameterization.

The model is forced with monthly surface water flux, daily shortwave and longwave radiation, and 6-hourly resolution air temperature, sea level pressure, humidity, and winds from the European Center for Medium Range Weather Forecast (ECMWF) ERA-Interim reanalysis product (0.75° resolution, Dee et al., 2011). River runoff from 54 river sources (35 in the



US) is explicitly represented. Daily water discharges from US rivers were retrieved from the US Geological Survey (USGS) river gauges (https://waterdata.usgs.gov). Climatologies from Mexican river discharges were derived from He et al. (2011), Munoz-Salinas and Castillo (2015), and Martinez-Lopez and Zavala-Hidalgo (2009). Monthly observations of dissolved inorganic nutrients (nitrate, ammonia, silicate) and organic nitrogen in the MS-A Rivers were retrieved from the USGS

(http://toxics.usgs.gov; Aulenbach *et al.*, 2007). Following Yu *et al.* (2015), the MS-A particulate organic nitrogen (PON) was determined as the difference between unfiltered and filtered total Kjendahl nitrogen (TKN), while the dissolved organic nitrogen (DON) was estimated as the difference between filtered TKN and ammonia. Only 10% of the estimated DON was incorporated into the model as labile DON, considering that most of the observed MS-A DON corresponds to refractory material (Green et al., 2006). Riverine PON was assigned to the small detritus pool. For river sources other than the MS-A,

dissolved inorganic nutrients and organic nitrogen concentrations are prescribed as climatological averages (USGS; Dunn, 1996; He *et al.*, 2011; Livingstone, 2015). Because Submarine Groundwater Discharge (SGD) is a significant source of nitrogen off the west Florida shelf (Hu et al., 2006), we included SGD-$NH_4$ fluxes based on rates reported by Swarzenski et al. (2007). We assumed that SGD-$NH_4$ fluxes occurred in regions shallower than 30 m, decreasing exponentially from 0.694 mmol m$^{-2}$ day$^{-1}$ at 10 m (minimum model depth) to 0.069 mmol m$^{-2}$ day$^{-1}$ at 30 m. Surface photosynthetically active

radiation (PAR) is assumed to be 43% of the surface shortwave radiation. Light attenuation includes a salinity dependent coefficient ($K_{salt}$) as in Fennel et al. (2011).

A 40-year model spin-up was completed before starting the historical simulation. Boundary conditions and surface fluxes for the model spin-up in each model year were extracted from a randomly selected year from the period 1979-2014, following Lee et al. (2011). After spin-up, the model was run continuously from January 1979 until December of 2014, with

20 monthly averaged fields saved.

## 3 Results

### 3.1 Model-data comparison

Modeled surface chlorophyll reproduced reasonably well spatiotemporal patterns in satellite chlorophyll (Fig. 2). The main differences between model and satellite chlorophyll are in the coastal region. Those differences can be explained (in

part) by satellite chlorophyll overestimation, due to the high concentration of dissolved colored organic matter and sediments associated with river runoff (Hu et al., 2000; Del Castillo et al., 2001; Gilbes et al., 2002; D'Sa and Miller, 2003). The greatest chlorophyll concentration values are within the Mississippi River delta, and the lowest values within the region influenced by the Loop Current. Significant seasonal differences are evident in the oceanic region, with minimum chlorophyll during summer, when thermal vertical stratification is the strongest, and maximum chlorophyll during winter,

concomitant with the greatest surface cooling and wind driven mixing (Muller-Karger et al., 1991; 2015). To compare temporal patters from model outputs and satellite observations, we derived monthly time series of chlorophyll in three regions: MS delta, Texas shelf, and a Deep Ocean area encompassing 86-92°W and 25-27.5°N (see regions in Figure 1). The




MS-delta and the Texas shelf are two productive regions strongly influenced by the MS-A river run-off, whereas the Deep Ocean box is an oligotrophic region often influenced by the Loop Current. Our model reproduces to a large extent the seasonal and interannual variability of satellite chlorophyll in the three regions (Fig. 3). Both model and satellite observations show maximum chlorophyll during March-May in the MS delta and Texas shelf, and during February-March in

the oceanic region. The average chlorophyll concentration of satellite chlorophyll is about 2.5-3 times greater than model chlorophyll in the MS delta and Texas shelf (Table 2). These differences are expected considering that satellite observations can be 2-4 times greater than in situ observations in the northern GoM, as reported in previous studies (Nababan et al., 2011). In the oceanic region, model chlorophyll overestimates SeaWiFS and MODIS chlorophyll by 7% and 14%, respectively, showing a significant similarity in term of amplitude and phase.

We evaluated the model's ability to reproduce interannual patterns of chlorophyll by performing Empirical Orthogonal Decomposition of chlorophyll anomaly time series (anomaly refers to monthly outputs/observations with the monthly climatological mean subtracted). The first EOF (EOF1) of model chlorophyll is consistent with the EOF1 from SeaWiFS (Fig. 4a, b) and MODIS (not shown), though the amplitude of the model chlorophyll mode is about half of the amplitude of the satellite chlorophyll modes. EOF1 is eminently a coastal pattern, with the greatest values located near the MS-A deltas.

The main differences between model and satellite EOF1 are located in the northwestern Florida region, where model chlorophyll is much lower than SeaWiFS chlorophyll, probably linked to a misrepresentation of the interannual variability in riverine nutrient load. The interannual variability of the first Principal Component (PC1) time series (which represents the temporal variability of EOF1) of model chlorophyll is well correlated to the PC1 time series of SeaWiFS (r = 0.69) and MODIS (0.60).

We compared the model-derived estimations of vertically integrated primary production with reported observations of phytoplankton production. The 1979-2014 average rates within the Louisiana shelf, Texas shelf, and open GoM (bottom depth > 1000 m) are 0.83, 0.37 and 0.24 g C m$^{-2}$ d$^{-1}$, respectively. Those estimates are in reasonable agreement with rates derived from in situ observations. Lohrenz et al. (2013) reported mean production values of 1.10 g C m$^{-2}$ d$^{-1}$ in the north-central shelf, 0.33 g C m$^{-2}$ d$^{-1}$ off Texas, and 0.28 g C m$^{-2}$ d$^{-1}$ in the open Gulf. The greatest disagreement with Lohrenz et al.

(2013) appears in the Louisiana shelf, where the model underestimates the mean production by 25%. Primary production patterns are strongly influenced by the MS-A river plumes, which are highly variable. Production values associated with the MS-A plumes exceed 1.5 g C m$^{-2}$ d$^{-1}$ (Fig. 5). Strong cross-shore gradients are observed during the spring in the Texas shelf, when prevailing easterly winds determine a westward and bottom advected MS-A river plume (Fig. 5a). Cross-shore gradient weakens during summer, when westerly winds promote up-coast flow that spreads the MS-A plume offshore (Fig.

5b). Distribution patterns over the Texas shelf are very consistent with the observations by Chen et al. (2000). They estimated maximum rates of ~1.6 g C m$^{-2}$ d$^{-1}$ near the MS-A delta region, and values <0.4 g C m$^{-2}$ d$^{-1}$ in the outer and western Texas shelves (see their Fig. 13).





## 3.2 Phytoplankton biomass patterns

As expected from the spatial pattern of chlorophyll and primary production, nanophytoplankton and diatom biomass show the greatest values near the MS-A delta and smallest in the deep GoM (Fig. 6a,b). Diatom concentration maxima are located west of the MS birdfoot delta, and near the mouth of the Atchafalaya River. In contrast, nanophytoplankton peaks near the Atchafalaya Bay, and dominates phytoplankton biomass over the Texas shelf. The diatom to total phytoplankton biomass ratios varied within ~0.35-0.6 on the Louisiana-Texas shelf (Fig. 6c), which is a reasonable range considering that the observed diatom ratios vary within ~0.15-0.6 (Zhao and Quigg, 2014).

Seasonal patterns in plankton biomass have important regional differences. To illustrate this, we estimated monthly climatologies of phytoplankton concentration from the surface to 30 m depth (or bottom depth if <30 m) within the MS delta, Texas shelf, and Deep Ocean regions (Fig. 7a-c; regions depicted in Fig. 1). Total phytoplankton is the greatest during April in the MS delta, March-April in the Texas shelf, and February-March in the oceanic region, and smallest during August in the three regions. The timing and amplitude of the seasonal maxima differ significantly between phytoplankton components. In the MS delta, diatoms peak in March, driving the phytoplankton biomass increase during late winter - early spring, whereas nanophytoplankton peak in April-May. The model derived diatom seasonality agrees with observations indicating greater diatom dominance during spring (Nelson and Dortch, 1996; Zhao and Quigg, 2014). In the Texas shelf, the spring phytoplankton maximum is mainly driven by nanophytoplankton. Diatoms do not have a marked spring peak like in the MS delta, displaying two maxima in February (the greatest) and June. In the oceanic region, nanophytoplankton drives the phytoplankton annual cycle, representing >90% of the total biomass, whereas diatoms do not have clear seasonality, displaying slightly smallest values during winter.

In the following sections we explore the underlying factors modulating spatio temporal changes in diatom and nanophytoplankton biomass. To this effect, we examine the driving factors of the specific growth rate variability, and investigate the changes in biomass production and losses.

## 3.3 Limitation factors and growth

To investigate the drivers of phytoplankton growth variability, we derive climatological patterns for the nutrient limitation factors ($L_P$; equations A1.5 and A2.7), the light limitation factors ($f_P$; equations A1.6 and A2.8), the temperature-dependent growth rates ($V_p$; equations A1.3 and A2.3), and the specific growth rate (SGR, which is the product of $L_P$, $f_P$ and $V_p$). We focus first on the mean spatial distribution of nutrient limitation, and then we analyze the seasonal patterns in nutrient limitation, light limitation, temperature-dependent growth, and specific growth rate. It is important to note that the nutrient and light limitation factors ranges from 0 to 1, with 0 indicating non-growth and 1 indicating no limitation. This implies that growth limitation is inversely related to the limitation factors.

Mean spatial patterns for the nutrient limitation factor from diatom ($L_{PL}$) and nanophytoplankton ($L_{PS}$) are presented in Fig. 8. Since $L_{PL}$ is defined as the minimum value between nitrogen limitation factor (NLF) and silica limitation factor



(SLF), we also present the SLF:NLF ratio to identify regions where diatoms are silica limited (SLF:NLF<1) and nitrogen limited (SLF:NLF>1) (black contours in Fig. 8a). Nutrient limitation in diatoms markedly increases away from the MS-A delta, with $L_{PL}$ varying from ~0.9 near the Atchafalaya Bay to <0.2 in the oceanic region (Fig. 8a). The SLF:NLF ratio indicates that diatoms are mainly N-limited in the northwestern and northeastern shelves, and Si-limited in the oceanic and

delta regions (Fig. 8a). Nanophytoplankton is much less nutrient limited than diatoms. $L_{PS}$ varies from ~1 in the Louisiana shelf to 0.35 in the deep GoM (Fig. 8b), while the $L_{PS}/L_{PL}$ ratio ranges from ~1.5 to >10 in the deep GoM (not shown). This explains the pronounced offshore decline in the diatom contribution to total phytoplankton (Fig. 6c).

    Seasonal changes in $L_{PS}$ and $L_{PL}$ for the MS delta, Texas shelf, and Deep Ocean are depicted in Fig. 9a, b. In the MS delta and Texas shelf, the nutrient-limitation factors are the greatest (i.e., the weakest limitation) during late winter and

spring, and the smallest (i.e., the strongest limitation) during fall, reflecting the seasonality in river discharge along the northern shelf (the maximum river discharge in Louisiana and Texas is during April and March, respectively, and the minimum in August-September). A secondary peak in the nutrient limitation factors is observed during summer in the Texas shelf, which can be related to wind-driven upwelling and a secondary peak in river discharge during summer. In the Deep Ocean, the nutrient-limitation factors are maxima during winter and minima during summer (though $L_{PL}$ seasonality in the

Deep Ocean is very weak), a pattern associated with the seasonal cycle in thermal stratification and mixing (enhanced mixed in winter, enhanced stratification in summer). Significant differences exist between the magnitude of $L_{PS}$ and $L_{PL}$. The $L_{PS}/L_{PL}$ ratio is ~1.5 in the MS delta, ~1.9 in the Texas shelf, and ~6-15 in the deep GoM. Unlike nanophytoplankton, diatoms can be considerably nutrient limited in the MS delta region. The SLF:NLF ratio (Fig. 7c) indicates that diatoms are mostly nitrogen limited in the Texas shelf, and silica limited in the Deep Ocean. In the MS delta, diatoms are more silica

limited during January-April, and more nitrogen limited during May-December. Still, the SLF:NLF ratio is close to 1 during December-June, indicating that both silica and nitrogen limitation can be important.

    Besides nutrients, light and temperature influence phytoplankton growth. The strongest light limitation is in the MS delta, and the weakest is in the deep GoM (Fig. 9d-e), but the regional differences in light-limitation are much smaller than those for nutrient-limitation. Seasonally, light limitation is weakest during April in the coastal regions, and May in the Deep

Ocean. Conversely, light limitation is strongest during August and December-January in the coastal regions, and December-January in the Deep Ocean. In the coastal regions, the decline in light limitation during summer can be linked to increased light attenuation, driven by the offshore spread of low-salinity and phytoplankton-rich waters by wind-driven upwelling. The temperature-dependent growth rate ($V_p$) displays the largest amplitude in the coastal regions, with a maximum in August and minimum in January-February (Fig. 9f-g). The ratio between the maximum and minimum $V_p$ is ~2.3 in the coastal regions

and ~1.4 in the Deep Ocean.

    The interplay among nutrient, light and temperature conditions determines the phytoplankton specific growth rate (SGR). The seasonal pattern in the SGR shows differences between coastal and oceanic domains (Fig. 9h-i). In the coastal regions, the inverse relationship between $V_p$ and both light and nutrient limitation factors during spring-summer determines the greatest SGR in June, while the small $V_p$ and light limitation factors during winter determine the minimum SGR in





December-January. In the Deep Ocean region, the SGR seasonality for nanophytoplankton (diatoms do not show a clear seasonal pattern) is mainly driven by nutrient and light limitation. The maximum SGR is in March, resulting from a trade-off between nutrient and light conditions during winter-spring, and the minimum SGR is in June-August, driven by the strong nutrient limitation during summer.

**3.4 Biomass sources and losses**

Now we explore how the patterns in phytoplankton production and losses influence the patterns in phytoplankton biomass. We showed that the SGR is the maximum during June in the coastal regions, and March in the Deep Ocean (Fig. 9h-i). We may expect that the seasonal changes in production reflect the changes in SGR, since production is the product between SGR and phytoplankton biomass. The link between SGR and production is evident in the Deep Ocean, as SGR and production

have maxima in March and minima during summer (Fig. 10c). However, in the MS delta and Texas shelf, the production peaks occur 1-3 month earlier than the SGR peaks (Fig. 10a, b). This necessarily implies that biomass losses due to biological (grazing, mortality, exudation) and physical (advection/diffusion) processes play an important role modulating production seasonality during spring-summer.

To evaluate how biologically driven processes influence the seasonal patterns in phytoplankton biomass, we calculated

the balance between production and biological losses (hereinafter the 'biological term'; Fig. 10d-f). The biological term displays distinct patterns for each phytoplankton component and region. The maximum biological term for diatom is in February in the MS delta, and January in the Texas shelf (no peak for diatoms in the Deep Ocean), while the maximum biological term for nanophytoplankton is in April in the MS delta, and February in the Texas shelf and Deep Ocean. The biological term for diatoms and nanophytoplankton begins to decline before the production maximum. Moreover, in the

Texas shelf, the biological term is negative during the production maximum. In the three regions, the biological term for total phytoplankton (diatoms plus nanophytoplankton) is positive in fall-winter, has a marked decline in spring, and is negative in summer. The seasonality of the biological term contrasts with the pattern in the SGR in the MS delta and Texas shelf, as SGR is minimum in winter and maximum in summer. All these features suggest that the seasonal changes in phytoplankton biomass are strongly modulated by biological losses. Zooplankton grazing is the dominant biological loss term (Fig. 10g-i),

markedly prevailing upon mortality and exudation (not shown). Microzooplankton exert the strongest grazing pressure on nanophytoplankton biomass, and mesozooplankton on diatoms, with the grazing patterns closely following the patterns in production. The seasonal patterns for microzooplankton (mesozooplankton) grazing upon nanophytoplankton (diatoms) closely follow the patterns in nanophytoplankton (diatom) production. Peaks in micro- and mesozooplankton grazing are concomitant or lag by 1 month the peak in nanophytoplankton and diatom production.

The seasonal patterns in the biological term do not completely explain the seasonal changes in phytoplankton biomass. To fully elucidate the net phytoplankton biomass change, the role of advection and diffusion as driver of biomass fluxes need to be examined. To this effect, we estimate the variability term representing advection and diffusion of phytoplankton biomass, hereinafter the physical term, and compare it with the biological term (Fig. 11a-c). The balance between these two



terms determines the net phytoplankton growth change. The physical and biological terms are generally inversely related, implying that the biologically driven changes tend to be offset by the physically driven changes. Besides, the biological term is generally larger than the physical term, and consequently the sign of the net phytoplankton growth is mainly determined by the biological component. The few exceptions are the negative growth during May in the MS delta, October in the Texas

shelf, and April and November in the Deep Ocean region. In these cases the physical term not only influence the amplitude of the monthly biomass change but the timing of the seasonal maxima. In the MS delta, the greatest magnitude for the physical term is during January-April, representing biomass losses mostly linked to horizontal advection (Fig. 11d). The advection can be related to the downstream export of phytoplankton rich water associated with the MS river plume. A substantial fraction of phytoplankton biomass from the MS-A delta is transported to the Texas shelf, which explains the

positive physical term during March-June (Fig. 11b,e). In the Deep Ocean, the greatest magnitude for the physical term is in November-March, representing biomass losses mainly driven by vertical diffusion (Fig. 11c,f). The close similitude between the physical and biological terms magnitude determines a much smaller net growth in the Deep Ocean than in the coastal regions (about 1 order of magnitude).

## 4 Discussion

We configured a 14-component model that explicitly considers two types of phytoplankton and zooplankton, and nitrogen and silica as limiting nutrients for phytoplankton growth. Inclusion of two phytoplankton components allowed for a realistic representation of the cross-shore gradients in biomass and chlorophyll. The model chlorophyll reproduced the temporal variability without significant seasonal bias. The good agreement between the model outputs and observations of chlorophyll and primary production provides confidence in the model's ability to represent seasonal phytoplankton dynamics. Our model

can be used to further advance ecological modeling capabilities, such as ecosystem responses to natural and anthropogenic disturbances in the GoM. We recognize that additional components and processes could be included in the model, such as phosphorus cycling and nitrogen fixation, to represent more realistic biogeochemical dynamics. Nevertheless, we believe that the current model configuration captures well enough the seasonal dynamics of diatoms and nanophytoplankton biomass in the GoM, which is the main goal in the present study.

We examined the main phytoplankton biomass patterns and explored the underlying factors explaining biomass variability following a similar approach to that used by Fennel et al. (2011). We used a constant depth layer (0-30 m), whereas Fennel et al. (2011) calculated seasonal patterns in a seasonally variable mixed layer depth (~10 m in summer to ~40 m in winter). We chose a constant depth layer because it makes the biomass budget analysis more straightforward. It is also worthwhile to mention that an important fraction of primary production can be distributed below the mixed layer in

spring-summer (Yu et al., 2015). The seasonal cycle for phytoplankton biomass in the 0-30 m layer differs in some degree from the seasonal cycle at surface (or in the mixed layer), with the latter showing a less pronounced biomass decline during





late spring in the coastal regions (Fig. 12). Consequently, the surface biomass changes do not necessarily reflect the biomass changes in the upper 30 m, but rather concentration changes in the seasonally variable mixed layer.

Our growth limitation analysis compared distinct regions in terms of phytoplankton production and river runoff influence, including the oligotrophic deep GoM, a region that has received less attention in previous modeling studies.

Nutrient limitation displayed the largest spatial differences compared to the other limiting factors (light and temperature), especially for diatoms. Diatoms are strongly silica limited in the deep GoM, nitrogen limited in most of the northern shelf, and both silica and nutrient-limited near the MS-A deltas. In the MS-A deltas, silicate limitation prevails during January-April, and nitrogen limitation during July-November, which agrees well with observations of severe silica depletion during spring, and nitrogen limitation during summer (Dortch and Whitledge, 1992; Nelson and Dortch,1996). Although

observational studies suggested the occurrence of silica limitation in the MS delta decades ago with a potential link to anthropogenic-driven declines in the MS river Si:N ratio (Turner and Rabalais, 1991), this is the first modeling study to evaluate the role of silica as driver of diatom growth in the region. The implication for silica and nitrogen limitation in the Louisiana-Texas shelf is that changes in the MS-A river nutrient load can modulate changes in diatom production, influencing phytoplankton composition. Changes in phytoplankton composition can have important repercussion for the

ecosystem, including changes in upper trophic levels dynamics and carbon export (carbon export is enhanced in diatom-dominated food webs). The latter may influence bottom hypoxia variability, as suggest by Dagg et al. (2003) and Green et al. (2008), and is an aspect that needs to be addressed in a future modeling study.

The SGR patterns showed important difference between coastal and oceanic domains. Nutrients, light, and temperature are important in modulating the seasonal SGR changes on the northern shelf, while nutrients and light are the dominant

factors driving the SGR seasonality in the deep GoM. The monthly averages for the SGR in small and large phytoplankton range within 0.32-0.85 and 0.22-0.60 day$^{-1}$ in the coastal regions, with the maximum (minimum) values in summer (winter). These SGR values are within the observational range reported by Fahnestiel et al. (1995), and similar to model estimations by Fennel et al. (2011). In the oceanic region, the SGR range for nanophytoplankton is 0.19-0.38 day$^{-1}$, with the maximum (minimum) values in late winter and early spring (summer).

Consistent with Fennel et al. (2011), we found that zooplankton grazing plays a leading role modulating phytoplankton biomass seasonality. This is especially evident in the coastal regions, where the balance between production and biological losses gives negative values (biomass decrease) in summer, and positive values (biomass increase) in winter, i.e. opposite to the pattern in the SGR. The role of zooplankton grazing as driver of phytoplankton seasonality has received increased attention in recent years. Behrenfeld (2010) proposed that increasing mixed layer depth (and consequently zooplankton and

phytoplankton dilution) leads to decreased grazing pressure during winter, resulting in a net population increase. Behrenfeld (2010) developed the Dilution-Recoupling hypothesis to explain the spring algal blooms in the North Atlantic, emphasizing the role of the biomass loss terms as drivers of net phytoplankton growth. This view challenged the prevailing Critical Depth hypothesis, which emphasizes the role of stratification, nutrients, and light as drivers of SGR and net phytoplankton growth (Gran and Braarud, 1935; Sverdrup, 1953). Our seasonal patterns in production, grazing and phytoplankton net growth are

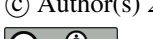



consistent with Behrenfeld's hypothesis, supporting the idea that phytoplankton seasonality is to an important degree top-down controlled.

Our study examined the coupled role of biologically (production and biological losses) and physically (advection and diffusion) driven biomass fluxes. Previous studies suggested the importance of advection and diffusion as driver of biomass changes in the GoM (e.g. Dagg et al., 2003; Green et al., 2008; Fennel et al., 2011). However, a quantification of these dynamics in biogeochemical model has not been done in the region. We found that the seasonal patterns in phytoplankton biomass are largely determined by small imbalances between biologically and physically driven fluxes, the latter mainly associated with horizontal advection in the Louisiana-Texas shelf, and vertical diffusion in the deep GoM. Consequently, we cannot obtain a proper understanding of biomass seasonality when the physically driven biomass fluxes are excluded from the analysis. Disentangling the processes influencing phytoplankton seasonality is a complex task, as the mechanisms acting as physical loss terms can also influence the balance between production and biological losses. That is the case for vertical diffusion, which modulates the vertical distribution of nutrients (impacting on phytoplankton production) and zooplankton (impacting on zooplankton grazing)(Behrenfeld, 2010).

Finally, future projections of environmental scenarios suggest substantial increases in both river runoff and thermal stratification in the northern GoM due to anthropogenic climate change (Tao et al., 2014; Liu et al., 2015). Therefore, how such environmental disturbances acting at multiple timescales can alter the subtle imbalances between primary production and biological losses (or between biological and physical driven biomass fluxes) is a topic that deserves further attention.

## 5 Summary and Conclusions

A coupled ocean-biogeochemical model was configured for the GoM to examine underlying mechanisms determining spatial and seasonal variability in diatoms and nanophytoplankton biomass. We investigated the factors modulating the specific growth rate (SGR), and explored the seasonal changes in biologically and physically driven biomass fluxes. We found that diatoms growth was ~40% and >95% nutrient-limited in the Louisiana shelf and deep GoM, respectively, whereas nanophytoplankton growth was ~10% and 40-80% limited. These differences strongly influenced the diatom contribution to total phytoplankton, which ranged from 50% near the MS delta to <10% in the deep GoM. Nutrient limitation for diatoms was mainly due to Si in the deep GoM, Si and N in the MS delta, and N elsewhere. The interplay among nutrient, light, and temperature determined the SGR seasonal timing (max/min) in the Louisiana-Texas shelf, while nutrient and light determined the SGR seasonal timing in the deep GoM. Primary production was driven by changes in SGR, but also influenced by biomass losses linked to zooplankton grazing. Moreover, the balance between primary production and biological losses revealed top-down control of phytoplankton growth. The physically driven biomass fluxes, mainly associated with horizontal advection in the Louisiana-Texas shelf and vertical diffusion in the deep GoM, played a key role modulating amplitude and phase in the seasonal phytoplankton biomass cycle. These results stress the importance of an




integrated analysis of biologically and physically driven biomass fluxes to better characterize phytoplankton biomass phenologies.

## Authors contributions

S-KL and FAG designed the study. FAG configured the model and performed the model simulations. FAG wrote the paper with contributions from all the authors.

## Conflicts of interests

The authors declare that they have no conflict of interest.

## Acknowledgments

We would like to thank Chris Kelble for his thoughtful comments and suggestion. This work was supported by the Northern Gulf Institute (NGI) and by the base funding of NOAA AOML

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



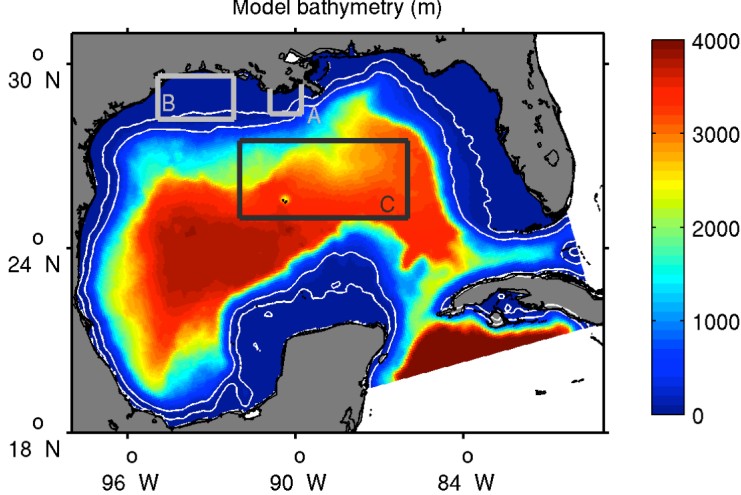

**Figure 1: Model domain and bathymetry. Polygons A, B, and C depict the MS delta, Texas shelf, and Deep Ocean region, respectively, selected to describe plankton patterns. White contours show the 50 and 200 m isobaths.**





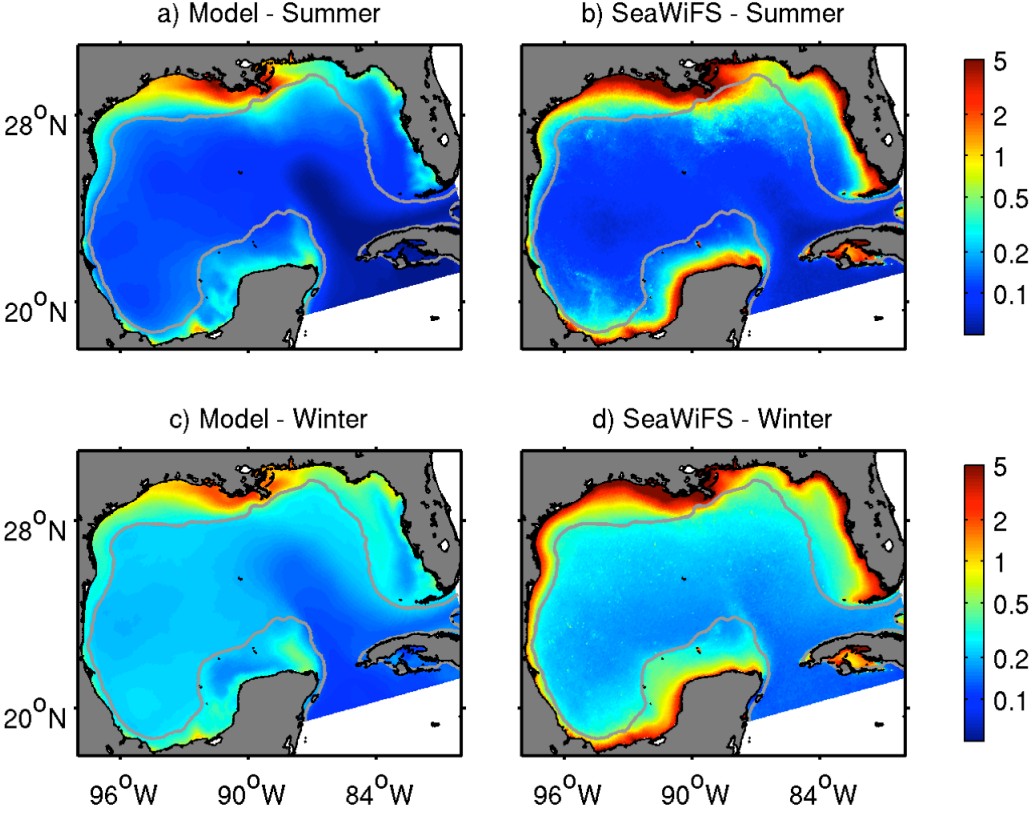

**Figure 2: Spatial patterns of model and satellite chlorophyll. Comparison between surface chlorophyll concentration (mg m$^{-3}$) derived from model outputs (a, c) and SeaWiFS (b, d) during summer and winter.**



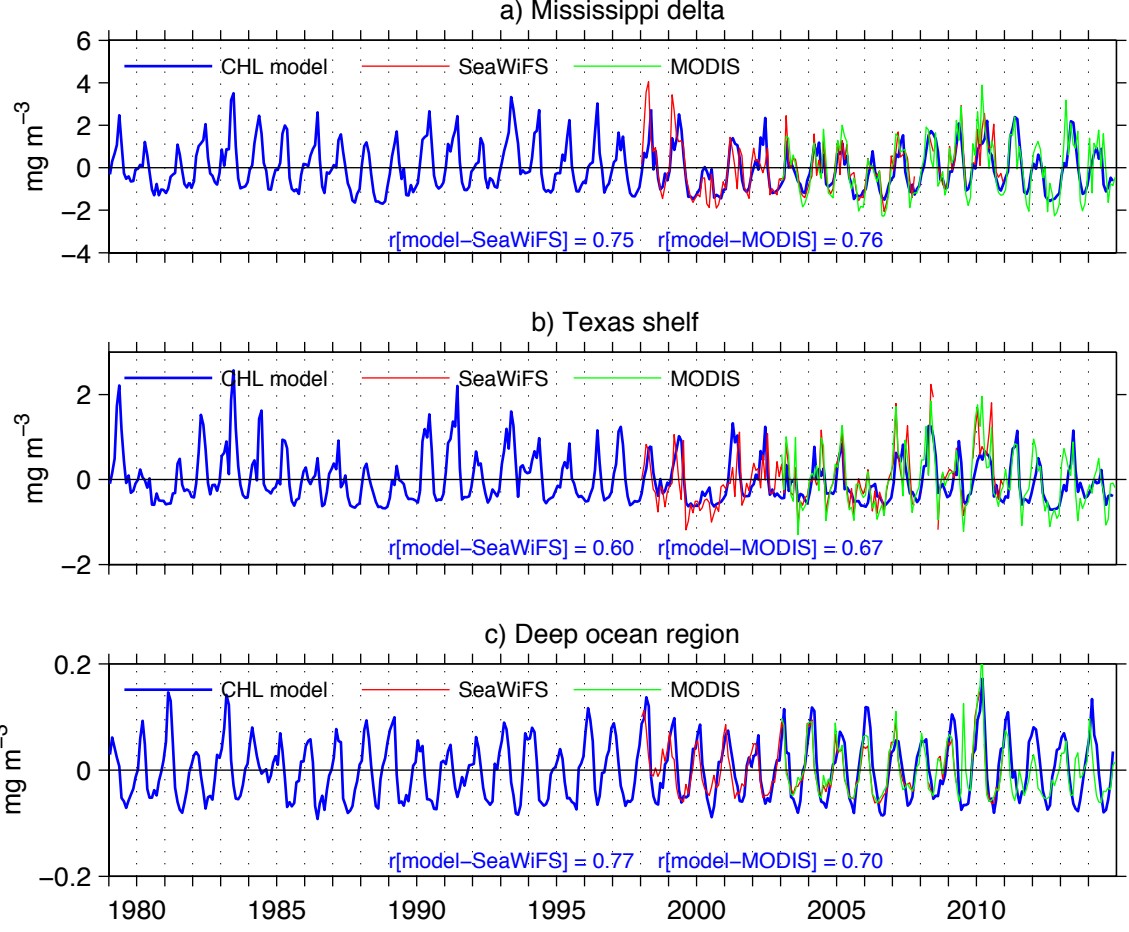

**Figure 3: Model and satellite chlorophyll time series. Long-term mean of each series was removed. Regions depicted in Fig. 1. Correlation coefficient between model and satellite time series is indicated at each panel.**



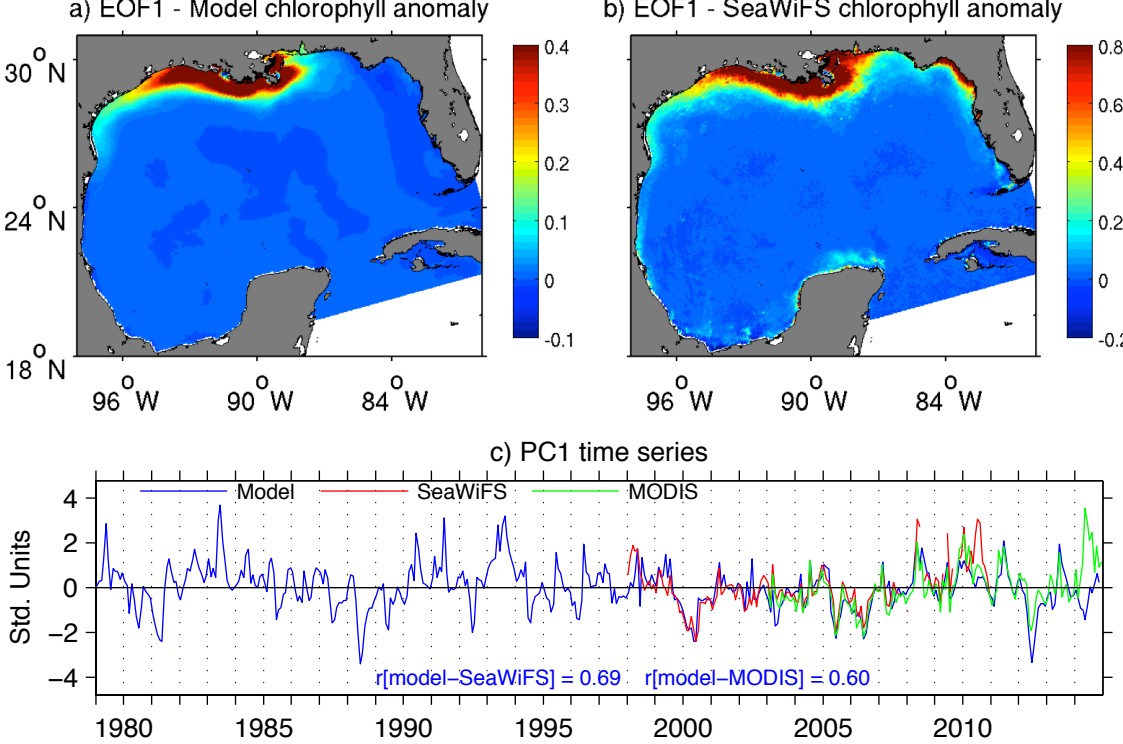

**Figure 4: EOF analysis of chlorophyll anomalies. a, b) First EOF mode of surface model chlorophyll (a) and SeaWiFS chlorophyll (b). c) Principal component associated with the first EOF mode of model, SeaWiFS, and MODIS chlorophyll. Correlation**
5 **coefficient between model and satellite PC1 series is indicated in panel c.**





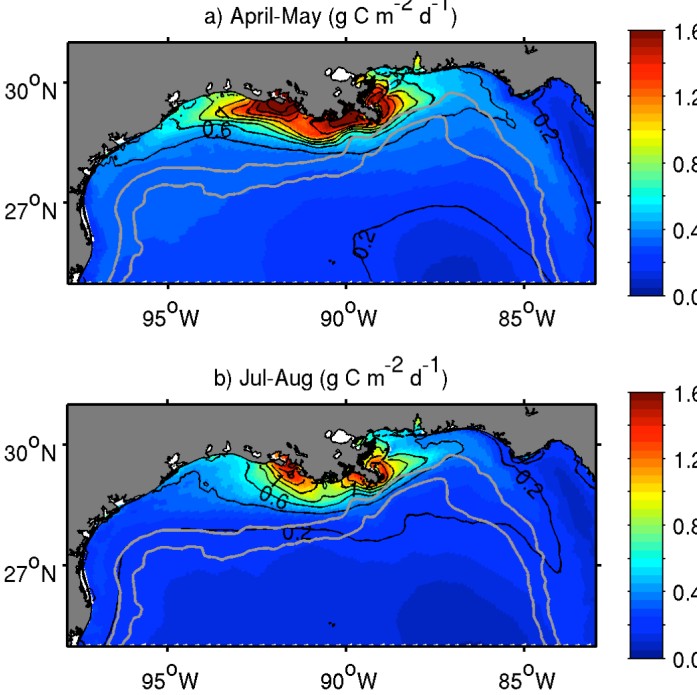

**Figure 5: Vertically integrated primary production patterns derived from biogeochemical model. Maps correspond to climatological averages for Apr-May (a) and Jul-Aug (b). Production contours (black) every 0.2 g C m⁻² d⁻¹. Gray contours depict the 200 and 1,000 m isobaths.**





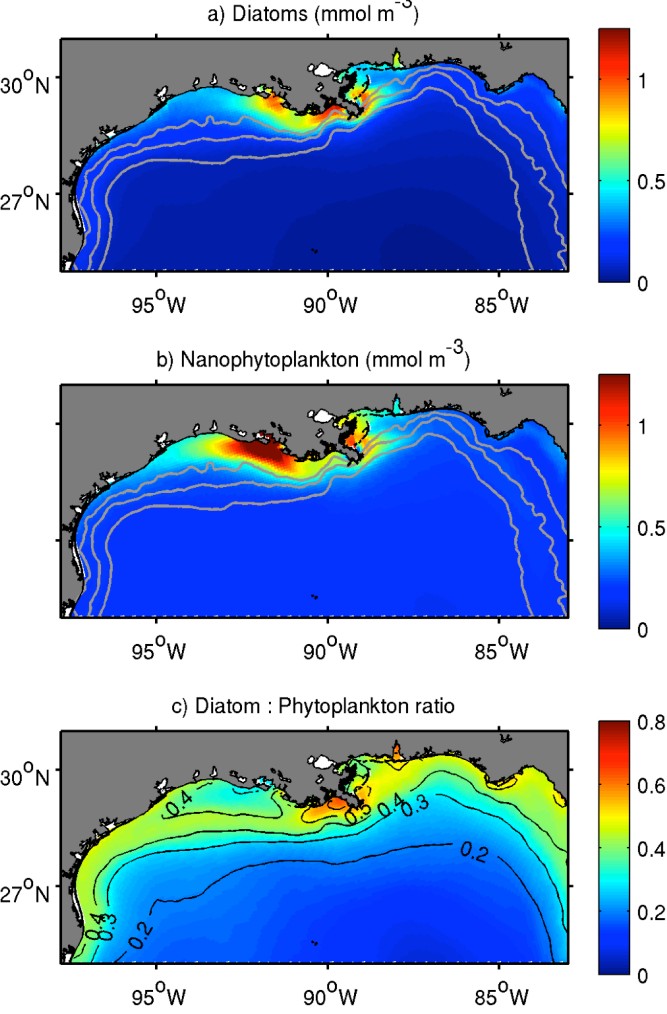

**Figure 6: Phytoplankton distribution. a-b) Mean surface fields of diatom and nanophytoplankton biomass; (c) Ratio of diatom to total phytoplankton biomass. Gray contours in panels a-b depict the 25, 50, and 200 m isobaths**




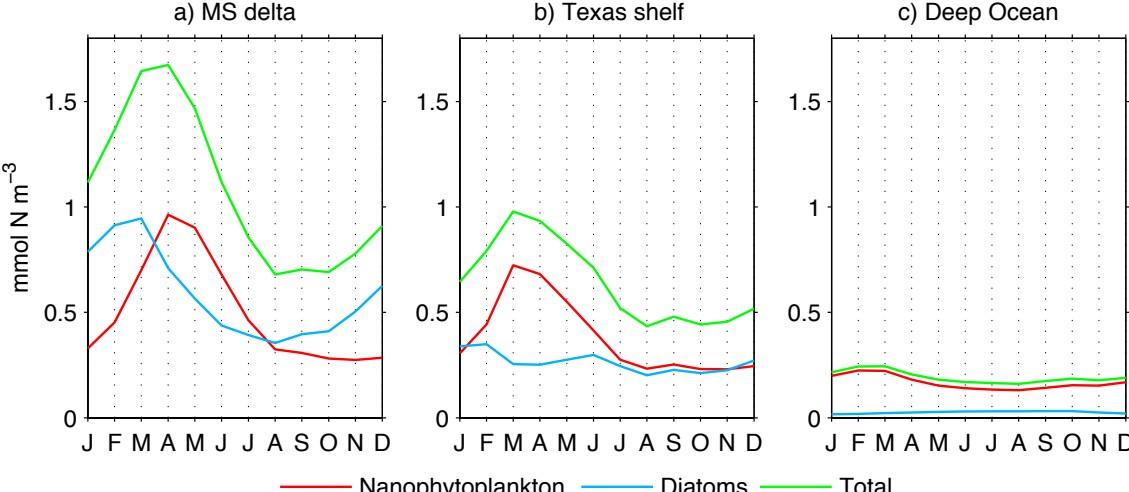

**Figure 7: Climatological seasonal cycle of phytoplankton biomass in the 30 m upper layer from the Mississippi delta, Texas shelf, and Deep Ocean (regions depicted in Fig 1, gray polygons).**





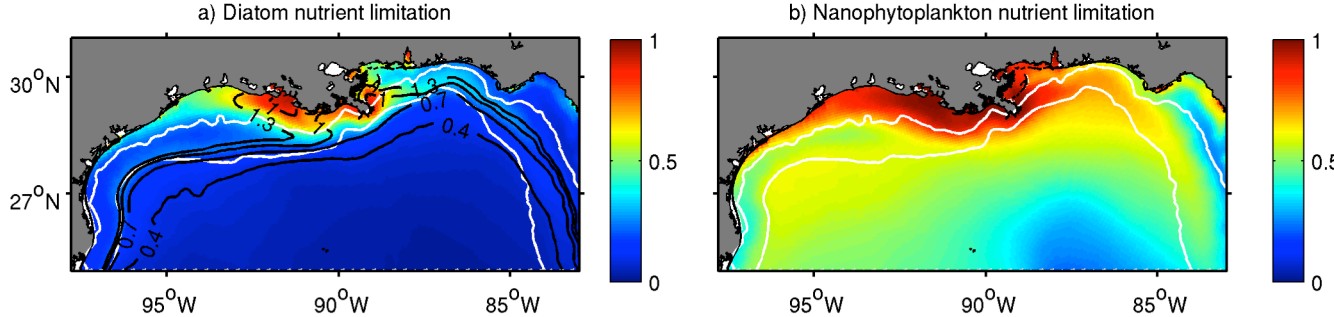

**Figure 8: Spatial patterns of nutrient limitation factors for diatoms (a) and nanophytoplankton (b). Nutrient limitation ranges from 1 (no limitation) to 0 (no growth). Black contours in panel a depict the ratio of silica to nitrogen limitation (SLF:NLF). SLF:NLF < 1 implies diatom's growth limitation is due to silica. White contours in panels a-d depict the 25 and 200 m isobaths.**



**Figure 9: Growth limitation and specific growth rates for nanophytoplankton (PS) and diatoms (PL): a-b) nutrient limitation factors; c) silica to nitrogen limitation ratio (SLF:NLF; for diatoms only); d-e) light limitation factors; f-g) temperature-dependent growth; h-i) specific growth rates. Factors were averaged in the upper 30 m layer from the Mississippi delta, Texas shelf and Deep Ocean regions (depicted in Fig 1, gray polygons).**



**Figure 10: Phytoplankton production, production minus losses (biological term), and grazing estimated for the upper 30 m layer of the Mississippi delta (left), Texas shelf (middle) and Deep Ocean (right) (regions depicted in Fig 1, gray polygons). Grazing terms are microzooplankton upon nanophytoplankton (PS2ZS) and diatoms (PL2ZS), and mesozooplankton upon nanophytoplankton (PS2ZL) and diatoms (PL2ZL).**

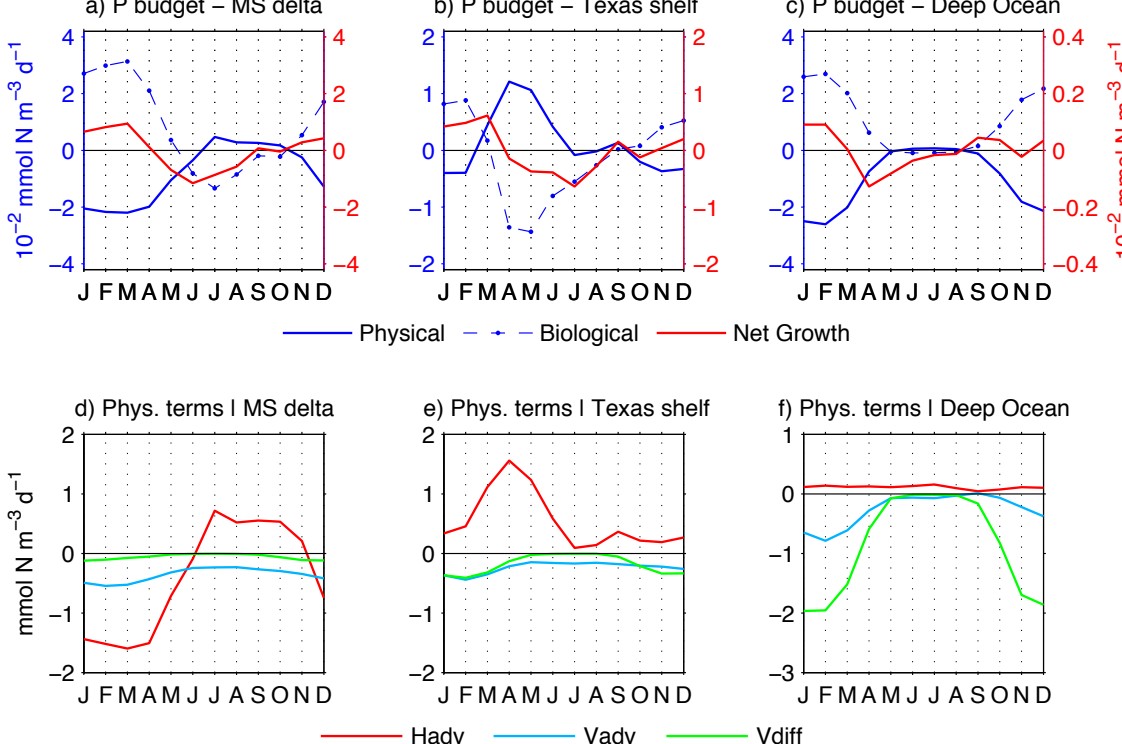

**Figure 11: a-c)** Phytoplankton biomass budget: the physical term represents advection plus diffusion, the biological term is production minus biological losses, and net growth (change rate) is the balance between the physical and biological terms. Right y-axis (red) is for net growth, and left y-axis (blue) is for the physical and biological terms. **d-f)** Physical terms components: Hadv, Vadv, and Vdiff correspond to horizontal advection, vertical advection, and vertical diffusion, respectively. Patterns are averages within the upper 30 m ocean layer from the Mississippi delta, Texas shelf and Deep Ocean box (regions depicted in Fig. 1).



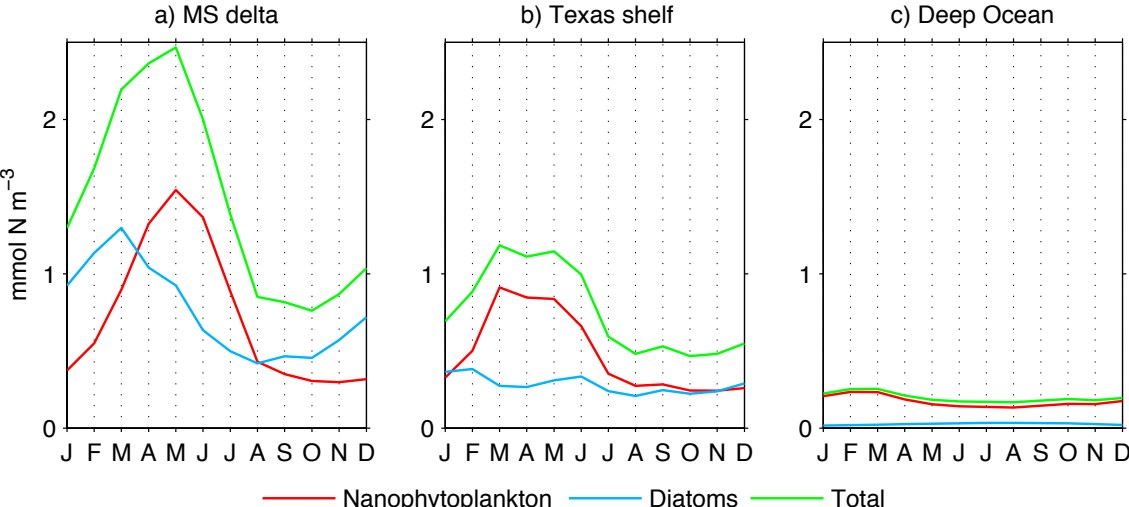

**Figure 12: As Fig. 7 but for surface phytoplankton.**



**Table 1. Model parameter values**

| Parameter | Name | | | Source |
|---|---|---|---|---|
| | Phytoplankton parameters | SP | LP | |
| $V_{max}$ | Maximum photosynthetic rate at 0°C (d$^{-1}$) | 0.51 | 0.76 | a, b, * |
| $k_{Gpp}$ | Temperature coefficient for photosynthesis (°C)$^{-1}$ | 0.0693 | 0.0642 | a |
| $\alpha_P$ | Initial slope of the P-I curve (m$^2$ W$^{-1}$) d$^{-1}$ | 0.028 | 0.030 | b |
| $KNO_3$ | Half saturation constant for nitrate (mmol N m$^{-3}$) | 1.0 | 3.0 | a |
| $KNH_4$ | Half saturation constant for ammonium (mmol N m$^{-3}$) | 0.1 | 0.5 | a |
| $KSi$ | Half saturation constant for silicate (mmol Si m$^{-3}$) | - | 3.0 | a |
| $\theta_{max}$ | Maximum chlorophyll to carbon ratio | 0.0268 | 0.0535 | b, c, * |
| $\varphi_P$ | Phytoplankton ratio extracellular excretion | 0.135 | 0.135 | a |
| PMor | Mortality at 0°C (m$^3$ mmolN$^{-1}$ d$^{-1}$) | 0.016 | 0.016 | * |
| $k_{PMor}$ | Temperature coefficient for mortality (°C)$^{-1}$ | 0.0693 | 0.0693 | a |
| $Att_P$ | Light attenuation due to chlorophyll (m$^2$ mg)$^{-1}$ | 0.0248 | 0.0248 | b |
| $w_P$ | Sinking rate (m day$^{-1}$) | - | 0.1 | b |
| | Zooplankton parameters | SZ | LZ | |
| $GR_{mPS}$ | Maximum grazing rate at 0°C on PS (d$^{-1}$) | 0.28 | 0.04 | d, * |
| $GR_{mPL}$ | Maximum grazing rate at 0°C on PL (d$^{-1}$) | 0.08 | 0.25 | d, * |
| $GR_{mZS}$ | Maximum grazing rate at 0°C on ZS (d$^{-1}$) | - | 0.15 | d, * |
| $k_{Gra}$ | Temperature coefficient for grazing (°C)$^{-1}$ | 0.0531 | 0.0531 | d |
| $K_{SPZ}$ | Half saturation on SP (mmol N m$^{-3}$)$^2$ | 0.20 | 0.85 | d, * |
| $K_{LPZ}$ | Half saturation on LP (mmol N m$^{-3}$)$^2$ | 0.20 | 0.85 | d, * |
| $K_{SZZ}$ | Half saturation on SZ (mmol N m$^{-3}$)$^2$ | | 0.85 | d, * |
| ZMor | Mortality at 0°C (m$^3$ mmolN$^{-1}$ d$^{-1}$) | 0.023 | 0.029 | * |
| $k_{ZMor}$ | Temperature coefficient for mortality (°C)$^{-1}$ | 0.0693 | 0.0693 | a |
| $\alpha_Z$ | Assimilation efficiency | 0.70 | 0.70 | a |
| $\beta_Z$ | Growth efficiency | 0.30 | 0.30 | a |
| | Detritus parameters | SD | LD | |
| $\tau_{NH4}$ | Decomposition to NH$_4$ rate at 25°C (d$^{-1}$) | 0.03 | 0.02 | b |
| $\tau_{DON}$ | Decomposition to DON rate at 25°C (d$^{-1}$) | 0.03 | 0.02 | b |
| $w_D$ | Sinking rate (m day$^{-1}$) | 1 | 10 | a, b, * |
| $k_D$ | Temperature coefficient for remineralization (°C)$^{-1}$ | 0.0693 | 0.0693 | a |

[a] Kishi et al. (2007); [b] Fennel et al. (2006; 2011); [c] Dune et al. (2010); [d] Gomez et al. (2017); [e] Yu et al. (2014), [f] Jiang et al. (2014); [g] Fennel et al. (2013); * Present study.



**Table 1 (Continuation)**

| Parameter | Name | Value | Ref. |
|---|---|---|---|
| Nit | Nitrification rate at 25°C ($d^{-1}$) | 0.05 | b |
| $k_{Nit}$ | Temperature coefficient for nitrification (°C)$^{-1}$ | 0.0693 | a |
| $I_{th}$ | Radiation threshold for nitrification inhibition (W m$^{-2}$) | 0.0095 | b |
| $D_p$ | Half-saturation radiation for nitrification inhibition (W m$^{-2}$) | 0.1 | b |
| $\gamma_{NH4}$ | DON decomposition to NH4 rate at 25°C ($d^{-1}$) | 0.04 | e, * |
| $\tau_{Si}$ | Opal dissolution to SiOH$_4$ rate at 25°C ($d^{-1}$) | 0.02 | f |
| $k_{DON}$ | Temperature coefficient for DON remineralization (°C)$^{-1}$ | 0.0693 | a |
| $k_{Si}$ | Temperature coefficient for opal dissolution (°C)$^{-1}$ | 0.0693 | a |
| $w_{Opal}$ | Opal sinking rate (m d$^{-1}$) | 10.0 | * |
| $Att_{sw}$ | Light attenuation due to seawater (m$^{-1}$) | 0.04 | b |
| Si:N | Silica to nitrogen ratio (mol Si (mol N)$^{-1}$) | 2.0 | a |
| C:N | Carbon to nitrogen ratio (mol C (mol N)$^{-1}$) | 6.625 | a, b |
| $R_{O2:NO3}$ | Oxygen to nitrate production ratio | 138/16 | g |
| $R_{O2:NH4}$ | Oxygen to ammonia production/consumption ratio | 106/16 | g |
| $R_{O2:Nitr}$ | Oxygen consumption during nitrification | 2.0 | g |
| $Ox_{th}$ | Oxygen threshold for aerobic respiration (mmol m$^{-3}$) | 6.0 | g |
| $K_{O2}$ | Oxygen half saturation for aerobic respiration (mmol m$^{-3}$) | 3.0 | g |



**Table 2. Long-term mean and standard deviation of model and satellite chlorophyll.**

|  | **MS delta** | **Texas shelf** | **Deep Ocean** |
|---|---|---|---|
|  | Mean (Std. Dev.) | Mean (Std. Dev.) | Mean (Std. Dev.) |
| Model: |  |  |  |
| 1979-2014 | 2.07 (1.08) | 0.96 (0.57) | 0.15 (0.06) |
| 1998-2010 | 1.98 (0.99) | 0.91 (0.49) | 0.16 (0.06) |
| 2003-2014 | 1.97 (0.99) | 0.87 (0.48) | 0.16 (0.06) |
| SeaWiFS: |  |  |  |
| 1998-2014 | 4.94 (1.19) | 2.41 (0.63) | 0.15 (0.05) |
| MODIS: |  |  |  |
| 2003-2014 | 5.61 (1.31) | 2.70 (0.67) | 0.14 (0.05) |
|  | Ratios | Ratios | Ratios |
| SeaWiFS/Model | 2.49 | 2.64 | 0.93 |
| MODIS/Model | 2.84 | 3.01 | 0.88 |