# Peer review of "Seasonal Patterns in Phytoplankton Biomass across the Northern and Deep Gulf of Mexico: A Numerical Model Study"

_Biogeosciences, 2017_

## Referee Comment (RC1) · K. Fennel (Referee) · 26 Dec 2017

General: This manuscript describes a new biogeochemical model for the Gulf of Mexico. The model results are analyzed with focus on seasonal variations in phytoplankton biomass, primary production, and relative abundance of diatoms versus small phytoplankton in the northern shelf region that is influenced by the Mississippi River and in the oligotrophic open Gulf. Previous regional modeling studies for the Gulf used simpler biogeochemical models that only include one phytoplankton functional group, so this study is a welcome extension. However, I have three major concerns that need to be addressed before I can recommend publication. These are related to model vali-

dation, the terminology used in describing model results, and a slight tendency by the authors to oversell their results while diminishing previous studies. These concerns are described in more detail below.

1) With regard to validation:

1.1) The authors provide no validation of the physical model. If there are previous publications in which this is reported, it would be fine to refer to those. Otherwise some physical model validation should be provided.

1.2) On page 6 (first paragraph) the authors state that the model underestimates mean satellite chlorophyll by factors between 2.5 to 3. These are rather large deviations in the mean. They then give reasons for why the satellite can't be trusted. There are two problems with this: first, the simpler models of Fennel et al. (2011) and Laurent et al. (2012) reproduced satellite chlorophyll without such large biases; and, second, if chlorophyll cannot be trusted it shouldn't be used in validation. However, satellite-derived chlorophyll is essentially the only data set that is used in this manuscript to validate the model.

1.3) There should be some validation of the biogeochemical model with in situ observations of phytoplankton biomass and nutrients. Such observations are available in the NODC and GOMRI databases. Profile comparisons for the open Gulf should be included.

1.4) Perhaps the largest omission, given the objective of the study, is that there is no validation of the different phytoplankton groups. I recognize that it is hard to get good data sets for this purpose, but there are some algorithms that can be used to separate satellite-derived chlorophyll into different size groups (see Hirata et al. 2011, Mouw et al. 2017 and references therein).

2) With regard to terminology: In section 3.4 the authors define the "biological term" as the balance between phytoplankton production and biological losses. This is the same

as Net Phytoplankton Growth, a widely used term in biological oceanography. It is not only unnecessary to redefine this as a new term, but also potentially confusing. Then the authors state that the balance of the biological and physical terms determines the change in net phytoplankton growth. This is wrong. Net phytoplankton growth is equal to what they defined as the biological term. The balance between the biological and physical terms is the local rate of change of phytoplankton.

3) In the following instances the authors should be more accurate in describing their results and the context in the existing literature:

3.1) In the last sentence of the abstract, they claim that their study shows the importance of representing large and small plankton in order to describe PP patterns. This is not supported by the results presented. On the one hand, there is no validation of the contributions of large and small phytoplankton to biomass and PP (see 1 above). On the other hand, there is no comparison to simulated phytoplankton abundance and PP from a model with only one phytoplankton group. Simpler models exist that, in fact, reproduce chlorophyll from satellite more accurately than this model in the Mississippi plume region (see comment 1.2).

3.2) In the second to fourth sentences of the discussion the authors make statements about their results that are not supported. "Inclusion of two phytoplankton components allowed for realistic representation ..." is not accurate as simple models arguably reproduced this better (see comment 1.2). "The good agreement between model outputs and observations of chlorophyll ..." is a questionable statement (see again comment 1.2).

3.3) With respect to phosphorus (P) the authors seem to be diminishing previous findings in an effort to justify why their model does not include P. On page 3 (line 8, sentence starting with "Although...") they seem to suggest that previous studies (specifically Laurent et al. 2012) suggest P limitation to be unimportant. This is not the conclusion of Laurent et al. (2012) nor of the follow-up studies by Laurent and Fennel (2014) and Fennel and Laurent (2017), which are consistent with the observational studies by Sylvan et al. (2006, 2007). All these studies do suggest the P limitation is critically important in the region influenced by the Mississippi River plume. Saying that P limitation is "moderate" while N and Si limitation are "critical" seems disingenuous. To be clear, I do not object to the fact that P is neglected in this model. All models are simplifications. It would be fine to state that their model neglects P, although it has been shown to be important in a portion of the model region. In the Discussion (end of first paragraph) it would be appropriate to be more forthcoming about previous studies on P limitation.

3.4) The statement in the Discussion (last sentence starting on page 11) about consistency with the dilution-recoupling hypothesis of Behrenfeld seems a bit cavalier. No detailed analysis in support of this statement was presented in this manuscript. The authors may want to consider the study by Kuhn et al. (2015), which used the same data set as Behrenfeld, and later papers by Behrenfeld where he backtracked himself somewhat from his early paper (Behrenfeld et al. 2013).

Other comments (not in order of importance):

4) P1, Line 13: Suggest inserting "improving" after "tools for"

5) P1, Line 14: Suggest removing "However"

6) P1, Line 19, sentence starting with "The model results show . . ." and following sentences in the abstract. Because diatoms in the model are strongly silica-limited doesn't necessarily mean they are in reality. Making inferences about reality from the model requires that the model accurately reproduces reality, which in this case is hard to prove. The authors certainly haven't (see my comments about validation). I would suggest that here and throughout the remainder of the abstract and manuscript the authors are more precise in their language. It is fine to say "diatoms in the model are silica limited" or some variation thereof. And "Simulated nanophytoplankton are . . ." rather than "Nanophytoplankton are . . ."

7) P1, Line 27: Suggest replacing "vertical diffusion" with "turbulent vertical diffusion" or "vertical mixing." Diffusion typically refers to molecular diffusion which acts on too small scales to make any difference to the processes considered here.

8) P1, Line 27, sentence starting with "This study highlights the . . ." This is an over-statement not supported by the results actually presented in this manuscript. See major comment 3.

9) P2, Line 9: ". . .because of deleterious impact on coastal ecosystems." The authors should provide one or more references in support of this statement, or modify it. I would like to challenge them to find a study that shows deleterious impacts on the ecosystem in the northern Gulf of Mexico (I am not aware of one). There are studies about specific aspects of the ecosystem, which would be fine to cite if sentence is slightly modified.

10) P3, Line 5, sentence beginning with "New modelling efforts . . ." I object to the logic of this statement. Adding complexity to biogeochemical models is not in itself a worth-while undertaking. It has to be motivated by the scientific questions (e.g. one might be interested in species succession). Sentence should be reformulated accordingly.

11) P3, Line 8: ". . .diatoms require . . ." Citing a modeling study (Kishi et al.) in support of a general statement about diatom traits seems inappropriate. There are more appro-priate references. I suggest the authors look up publications by Elena Litchman and collaborators. She has worked extensively on documenting phytoplankton functional traits.

12) P4, Line 30: Which basin does "basin-scale" refer to here?

13) P5, Line 18: ". . .randomly selected year" Which year?

14) P5, Line 23: Stating the model "reproduces" the observations is an overstatement. It would be more appropriate to say they agree qualitatively.

15) P5, Line 31: The authors should make it much more clear upfront that these are anomalies (i.e. that the bias was removed).

16) Results, general: No oxygen results are shown. Given this, there is not much point saying the model includes oxygen.

References

Behrenfeld, M.J., Doney, S.C., Lima, I., Boss, E., Siegel, D.A., 2013. Annual cyles of ecological disturbance and recovery underlying the subartic Atlantic spring plankton bloom. Glob. Biogeochem. Cycles 27, 526–540. doi:10.1002/gbc.20050.

Fennel, K. and Laurent, A.: N and P as ultimate and proximate limiting nutrients in the northern Gulf of Mexico: Implications for hypoxia reduction strategies, Biogeosciences Discuss., https://doi.org/10.5194/bg-2017-470, in review, 2017.

Kuhn, A.M., Fennel, K., Mattern, J.P., 2015. Model investigations of the North Atlantic spring bloom initiation. Prog. Oceanogr. 176–193.

Laurent, A., Fennel, K., Simulated reduction of hypoxia in the northern Gulf of Mexico due to phosphorus limitation, Elementa 2:000022, doi:10.12952/journal.elementa.000022 (2014)

Mouw et al. (2017) https://www.frontiersin.org/article/10.3389/fmars.2017.00041

Hirata, T., Hardman-Mountford, N. J., Brewin, R. J. W., Aiken, J., Barlow, R., Suzuki, K., Isada, T., Howell, E., Hashioka, T., Noguchi-Aita, M., and Yamanaka, Y.: Synoptic relationships between surface Chlorophyll-a and diagnostic pigments specific to phytoplankton functional types, Biogeosciences, 8, 311-327, https://doi.org/10.5194/bg-8-311-2011, 2011.

Sylvan JB, Dortch Q, Nelson DM, Brown AFM, Morrison W, Ammerman JW. 2006. Phosphorus limits phytoplankton growth on the Louisiana shelf during the period of hypoxia formation. Environ Sci Technol 40:7548–7553.

Sylvan JB, Quigg A, Tozzi S, Ammerman JW. 2007. Eutrophication-induced phosphorus limitation in the Mississippi River Plume: evidence from fast repetition rate fluorometry. Limnol Oceanogr 52:2679–2685.

---

## Referee Comment (RC2) · Anonymous Referee #2 · 28 Dec 2017

General comments:

This paper investigates the spatial and seasonal patterns of phytoplankton biomass in the Gulf of Mexico (GoM) using a three-dimensional biogeochemical model that could explicitly simulate small- and large-size plankton groups. The authors demonstrate that the model could reproduce the satellite observed dominant seasonal patterns of phytoplankton biomass in GoM and explore the underlying mechanisms controlling the seasonal variability. This work is of interest to the community and would complement previous modeling work to improve understanding of phytoplankton dynamics in GoM. However I have a few concerns, as listed below, but subject to these being addressed

[Figure]

I would recommend the paper for publication in Biogeosciences.

1. My major concern is associated with the validation of the coupled physical-biogeochemical model:

First, there is no physical validation presented in the paper, despite that the authors have emphasized the importance of physical processes on the net phytoplankton growth. Has the physical validation work been done and/or published elsewhere? If yes, it is important to summarize that here in some way. If not, I think it's worthwhile to do some extra work on physical validation to make the presented results here more convincing considering how important the physics is controlling the biogeochemical cycling in this region (e.g., the mixing and transport by riverine waters to northern GoM, Loop Current and eddy interactions to deep GoM, etc.). For example, the simulated spatial extent of the high chlorophyll river plume in northern GoM is narrower than that observed in satellite (visually viewed from Fig. 2), could it be associated with the distant transport of riverine nutrients?

Second, the validation of biogeochemical (BGC) model doesn't seem sufficient to me. The BGC validation in the paper primarily relies on comparing model simulated and satellite observed surface chlorophyll. While the model overall reproduces the dominant seasonal and spatial patterns in satellite chlorophyll, it significantly underestimates the coastal chlorophyll both in magnitude (2.5-3 times lower in the model) and spatial extent. The authors attribute the mismatch to satellite overestimating in situ observations of chlorophyll in northern GoM. If true, it would be useful to also include comparisons between simulated and in situ observations of chlorophyll in the paper for justification. In addition, while satellite chlorophyll observations have the advantage for model validation due to its spatial and temporal coverage, they are limited to the first optical depth that could hardly represent the plankton dynamics in subsurface water (e.g., the deep chlorophyll maxima). Hence a good complement to the validation might be including comparison to chlorophyll profiles, which to my knowledge is available in GoM during the model simulation period (e.g., the bio-optical profiling float results presented

in Green et al., 2014). Also, there are relatively 'abundant' observations, apart from chlorophyll, in the northern GoM, such as those provided by Mechanisms Controlling Hypoxia (MCH) program (http://hypoxia.tamu.edu/field-program), in situ observations of primary production (Lehrter et al. 2009), and water column community respiration rates (Murrell et al. 2013). These datasets might improve the BGC validation in coastal region where satellite chlorophyll is considered to have higher uncertainty.

2. One novelty of this work is that the model includes two phytoplankton types and two zooplankton types that complement the previous modeling work in GoM that mostly only includes one phytoplankton and one zooplankton type. While the additional complexity added to the BGC model is more faithful in representing the lower-trophic level dynamics in real system, it also adds more complexities and challenges in calibrating and validating the model. With respect to calibration, have the parameter values shown in Table 1 (especially those with *) been informally or formally tuned or optimized? Are the conclusions presented here sensitive to the selected parameter values? I think providing more information/comments on these would be helpful to others. The additional complexity of the BGC model also adds difficulties in model validation, e.g., the model-data chlorophyll comparison alone cannot tell how reasonable the model simulates each type of phytoplankton group as it could not distinguish the contribution from small- and large-size phytoplankton groups. How has the added complexity benefit us to understand the plankton dynamics in this region? Does the presented model do a better job than the previous modeling work that only include one phytoplankton type (e.g., compared with Xue et al. 2013)? I think readers would appreciate with a bit more discussions/comments on these.

Specific comments:

1. Page 4, Line 6: Would it be more appropriate to list an observational rather than a modeling work (Xue et al., 2013) as a reference?

2. Page 4, Line 14: delete one 'to' either in front of the ':' or after the number.

[Figure]

3. Page 4, Line 22: Why listed MODIS SST here? Has it been used anywhere in the paper?

4. Page 4, Line 28: Horizontal diffusivity is non-zero here, but it seemed to be neglected when analyzing the role of advection and diffusion in section 3.4.

5. Page 4, Line 30: Does the basin-scale model also include biogeochemistry and provide BGC initial conditions? If not, how do you specify them? Could you also provide more information on how you specify open boundary conditions? Has tide been included?

6. Page 5, Line 18: Where were the boundary conditions and surface fluxes extracted from? the basin-scale model?

7. Page 6, Line 23: 'mean production values', is it spatial or/and temporal mean? Maybe also provide the standard deviation if available, since the primary production is highly variable?

8. Page 7, Line 29: change 'ranges' to 'range'?

9. Page 8, Line 26: In the text, it's switching between 'summer' (or winter) and 'months' back and forth. Could you specify the summer and winter months at the first time they appear?

10. Page 10, Line 22-24: This statement is a bit exaggerated to me since the validation is on chlorophyll, a combination of two phytoplankton groups, that how well each type of phytoplankton is simulated by the model is not directly validated.

11. Fig.2: the lower limit of the color bar is missing? From 0? What does the gray contour line represent? 200m isobath?

12. Fig.8: should be '...in panels a-b depict ...'

References

Green, R.E., A. S. Bower, and A. Lugo-Fernandez (2014), First autonomous bio-optical profiling float in the Gulf of Mexico reveals dynamic biogeochemistry in deep waters. PLoS One, 9, 1–9. doi:10.1371/journal.pone.0101658

Lehrter, J. C., M. C. Murrell, and J. C. Kurtz (2009), Interactions between Mississippi River inputs, light, and phytoplankton biomass and phytoplankton production on the Louisiana continental shelf, Cont. Shelf Res., 29, 1861–1872, doi:10.1016/j.csr.2009.07.001.

Murrell, M. C., R. S. Stanley, J. C. Lehrter, and J. D. Hagy (2013), Plankton community respiration, net ecosystem metabolism, and oxygen dynamics on the Louisiana continental shelf: Implications for hypoxia, Cont. Shelf Res., 52, 27–38, doi:10.1016/j.csr.2012.10.010.
* * *

---

## Referee Comment (RC3) · Anonymous Referee #3 · 1 Jan 2018

The paper by Gomez et al presented a coupled physical-biogeochemical modeling study for the Gulf of Mexico. By combining the NEMURO NPZ model by Kishi et al., 2007 and the biogeochemical model by Fennel et al., 2011, the author tried to introduce a more complicated nutrient-low trophic level model to the Gulf of Mexico, which is plausible. Nevertheless, I totally agree with the major concerns from the other two reviewers, the study presented so far failed to demostrate their model's credibility for either the physical or biogeochemical part. Given these model validation been provided, the paper failed to prove the benefits of introducing this multi-plankton group lower trophic level model. My major concerns, alike those of the two other reviewers', are as follows:

[Figure]

1) Validation of the physical model. The paper stated that the boundary conditions were from a HYCOM model, yet the model (ROMS)'s own performance regarding circulation and T/S fields was not evaluated, without which, I would have a big question mark about the results presented in the manuscript; 2) Validation of the biogeochemical model. The author evaluated their model's performance via a comparison against satellite data and admitted that their model underestimated the Chl-a. And unfortunately, these satellite data were the only source used for model evaluation. How about the model's performance on nutrient and plankton groups? Without such information, it is hard to conclude that the model could at least represent the nutrient and biological cycle in the Gulf; 3) Given that point 1) and point 2) were addressed, I could not find the benefit of introducing the new plankton group (2 phytoplankton and 3 zooplankton vs. 1 phytoplankton and 1 zooplankton by Fennel at al. 2011), which, indeed, could be the most important contribution of this study.

---

## Author Comment (AC1) · 18 Feb 2018

We highly appreciate the reviewers for all their valuable comments and suggestions. We have included our responses in blue. References and Supplementary Figures (Fig. S) are after the response sections.

**1. Response to Dr. Fennel**

**1.1 Validation**

**1.1.1. Physical validation:**

The authors provide no validation of the physical model. If there are previous publications in which this is reported, it would be fine to refer to those. Otherwise some physical model validation should be provided.

We agree that the manuscript needs some physical-component validation. We have done comparisons that show a good agreement between model outputs and observation of SST, coastal sea level anomalies, eddy kinetic energy, and surface salinity (See Figs. S1 to S5). In addition to those variables, in the revised manuscript version we will provide validation for time series of salinity and vertical profiles of temperature-salinity, in both coastal and oceanic domains.

**1.1.2. Chlorophyll patterns:**

On page 6 (first paragraph) the authors state that the model underestimates mean satellite chlorophyll by factors between 2.5 to 3. These are rather large deviations in the mean. They then give reasons for why the satellite can't be trusted. There are two problems with this: first, the simpler models of Fennel et al. (2011) and Laurent et al. (2012) reproduced satellite chlorophyll without such large biases; and, second, if chlorophyll cannot be trusted it shouldn't be used in validation. However, satellite derived chlorophyll is essentially the only data set that is used in this manuscript to validate the model.

We understand the concern pointed out by the reviewer regarding to the large deviation in the mean of model and satellite chlorophyll in the coastal region. To support better our results, we will extend the validation analysis for chlorophyll, including a comparison between model outputs and in situ observations. We are still working in this validation analysis, but preliminary comparisons indicate that model chlorophyll tends to underestimate in situ chlorophyll in the Louisiana-Texas inner shelf (see Figures S6-S7). An estimation of the differences between model and in-situ chlorophyll will be reported in the revised manuscript version.

Regarding to the first related problem (a simpler model reproduces better the chlorophyll variability), we agree that the Fennel model matches better the chlorophyll pattern in the Louisiana-Texas shelf, as shown by Fennel et al. (2011) and Laurent et al. (2012). However, the Fennel model tends to overestimates satellite chlorophyll in the open ocean region, which can be noted in the chlorophyll patterns reported by Xue et al. (2013) [see their Figure 8]. To evaluate better to what degree the chlorophyll patterns from Fennel and our 18-component model (referred to as GOM) differ, we run the Fennel model for the period 1999-2004, using the same parameters values used by Fennel et al. (2011).

The comparison between model-derived and satellite chlorophyll patterns (Fig. S8 and S9) indicates that Fennel model overestimates satellite chlorophyll in most of the open ocean region (bottom depth >200 m). This model bias displays a seasonal pattern, with the strongest deviations occurring in winter and early spring (Dec-Mar) (the overestimation factor can be >3; Fig. S8c). On the other hand, the chlorophyll from the GOM model reproduces better the mean satellite condition and temporal variability in the open ocean region, showing no large seasonal deviation. In the Mississippi Delta and Texas shelf, there is a clear difference between the long-term mean of Fennel and GOM (Fennel matching well the SeaWiFS mean, Fig. S8a-b). However, the Fennel-SeaWiFS and GOM-SeaWiFS correlation coefficients are pretty similar (in terms of correlation, Fennel does better than GOM in the Texas shelf, and GOM does better than Fennel in the Mississippi Delta), indicating a similar model ability to reproduce the temporal variability of chlorophyll. We will add this comparison within the result section.

Regarding to the second problem (if chlorophyll cannot be trusted it shouldn't be used in validation), previous studies have shown that although satellite chlorophyll can largely overestimate in situ chlorophyll in regions influenced by river runoff (until by 300%), there is a significant correlation between satellite and in situ chlorophyll (as example, see Figure 1.10 in Nababan, 2005). Besides, the bias associated to optically complex waters occurs mainly in the coastal region but not in Open Ocean. These two aspects, added to the fact that satellite sensors provide large spatial and temporal coverage to properly describe seasonal and interannual variability, makes the model-satellite comparison of chlorophyll valid.

**1.1.3. Biomass and nutrients:**

There should be some validation of the biogeochemical model with in situ observations of phytoplankton biomass and nutrients. Such observations are available in the NODC and GOMRI databases. Profile comparisons for the open Gulf should be included.

We agree that the manuscript requires validation for the biological and chemical components. Therefore, a validation of nutrient and plankton biomass, using the NODC and GOMRI databases, will be provided.

**1.1.4. Diatom and nanophytoplankton:**

Perhaps the largest omission, given the objective of the study, is that there is no validation of the different phytoplankton groups. I recognize that it is hard to get good data sets for this purpose, but there are some algorithms that can be used to separate satellite-derived chlorophyll into different size groups (see Hirata et al. 2011, Mouw et al. 2017 and references therein).

As Dr. Fennel indicates, it is difficult to obtain data set to validate the diatom and nanophytoplankton components. We will not include a satellite-based analysis on phytoplankton group biomass pattern, as time limitations preclude performing this type of analysis (which we think can be the topic for an independent paper). Although biomass patterns of phytoplankton functional groups are not well documented in the region, few studies have reported data on size-fractioned chlorophyll, and phytoplankton group abundance/biomass (Bode and Dortch, 1996; Nelson and Dortch, 1996; Dagg and Breed, 2003; Zhao and Quigg, 2014) that can be compared with our model. Figure S10 shows model and observed diatom to total chlorophyll ratio for two coastal stations (A and B) over the Louisiana shelf, taken from data reported by Zhao and Quigg (2014). In station B, both model and observed chlorophyll ratio are similar. In station A, the model ratio overestimates the observed ratio, especially in August. Both simulated and observed ratios show greater values during April than during August. Though differences between model and observations are evident, the standard deviation bars indicate large variability, which can be linked to strong mesoscale variability in the Mississippi delta region (e.g. Marta-Almeida et al., 2013). We considered that the model does a reasonable work simulating those reported chlorophyll ratios, and is qualitatively consistent with the seasonal pattern. We will include Figure S10 in the validation section of the revised paper version.

**1.2.** Terminology**

With regard to terminology: In section 3.4 the authors define the "biological term" as the balance between phytoplankton production and biological losses. This is the same as Net Phytoplankton Growth, a widely used term in biological oceanography. It is not only unnecessary to redefine this as a new term, but also potentially confusing. Then the authors state that the balance of the biological and physical terms determines the change in net phytoplankton growth. This is wrong. Net phytoplankton growth is equal to what they defined as the biological term. The balance between the biological and physical terms is the local rate of change of phytoplankton.

We highly appreciate the term names clarification, and understand that is preferable avoid any confusing terminology in the budget analysis. We will modify the term names following Dr. Fennel indications.

**1.3.** Improving description of the results and the context in the existing literature:**

1.3.1. In the last sentence of the abstract, they claim that their study shows the importance of representing large and small plankton in order to describe PP patterns. This is not supported by the results presented. On the one hand, there is no validation of the contributions of large and small phytoplankton to biomass and PP (see 1 above). On the other hand, there is no comparison to simulated phytoplankton abundance and PP from a model with only one phytoplankton group. Simpler models exist that, in fact, reproduce chlorophyll from satellite more accurately than this model in the Mississippi plume region (see comment 1.2).

We agree that that sentence is not supported by the result presented and will be removed from the abstract.

1.3.2. In the second to fourth sentences of the discussion the authors make statements about their results that are not supported. "Inclusion of two phytoplankton components allowed for realistic representation..." is not accurate as simple models arguably

reproduced this better (see comment 1.2). "The good agreement between model outputs and observations of chlorophyll..." is a questionable statement (see again comment 1.2).

We recognize that Fennel model match better the satellite chlorophyll patterns than our model in the coastal regions, and that we need to tone down those statements. However, it is worth to note that our model reproduces better than Fennel model the satellite chlorophyll pattern in the deep ocean region (see answer to comment 1.1.2), so we will comment about this in the discussion.

1.3.3. With respect to phosphorus (P) the authors seem to be diminishing previous findings in an effort to justify why their model does not include P. On page 3 (line 8, sentence starting with "Although: : :") they seem to suggest that previous studies (specifically Laurent et al. 2012) suggest P limitation to be unimportant. This is not the conclusion of Laurent et al. (2012) nor of the follow-up studies by Laurent and Fennel (2014) and Fennel and Laurent (2017), which are consistent with the observational studies by Sylvan et al. (2006, 2007). All these studies do suggest the P limitation is critically important in the region influenced by the Mississippi River plume. Saying that P limitation is "moderate" while N and Si limitation are "critical" seems disingenuous. To be clear, I do not object to the fact that P is neglected in this model. All models are simplifications. It would be fine to state that their model neglects P, although it has been shown to be important in a portion of the model region. In the Discussion (end of first paragraph) it would be appropriate to be more forthcoming about previous studies on P limitation.

We understand the point indicated by the reviewer and recognize that phosphorous limitation can be critical in the region influenced by the Mississippi River. We will modify the statement in the 'Model description' section to:

"Although previous modeling studies have indicated the existence of phosphate limitation near the MS-A deltas during May-July (Sylvan et al. 2006, 2007; Laurent et al., 2012; Laurent and Fennel, 2014), we focus here on the role of N and Si, as observational studies suggest that N and Si can modulate phytoplankton composition (Dortch and Whitledge, 1992; Nelson and Dortch, 1996; Lohrenz et al., 1997; 2008; Rabalais et al., 2002; Zhao and Quigg, 2014)."

**In addition, we will be more forthcoming about previous studies on P limitation, incorporating within the discussion the studies by Laurent et al. (2012), Laurent and Fennel (2014), Fennel and Laurent (2017), and Sylvan et al. (2006, 2007).**

1.3.4. The statement in the Discussion (last sentence starting on page 11) about consistency with the dilution-recoupling hypothesis of Behrenfeld seems a bit cavalier. No detailed analysis in support of this statement was presented in this manuscript. The authors may want to consider the study by Kuhn et al. (2015), which used the same data set as Behrenfeld, and later papers by Behrenfeld where he backtracked himself somewhat from his early paper (Behrenfeld et al. 2013).

We agree with the reviewer that more analyses need to be done to support the consistency between the dilution-recoupling hypothesis and our model results. Since this aspect is beyond the paper goals, we decided to remove the dilution-recoupling hypothesis part from the discussion.

**Other comments (not in order of importance):**

1.4. P1, Line 13: Suggest inserting "improving" after "tools for"

The change will be done accordingly.

**1.5.** P1, Line 14: Suggest removing "However"

The change will be done accordingly.

**1.6.** P1, Line 19, sentence starting with "The model results show..." and following sentences in the abstract. Because diatoms in the model are strongly silica-limited doesn't necessarily mean they are in reality. Making inferences about reality from the model requires that the model accurately reproduces reality, which in this case is hard to prove. The authors certainly haven't (see my comments about validation). I would suggest that here and throughout the remainder of the abstract and manuscript the authors are more precise in their language. It is fine to say "diatoms in the model are silica limited" or some variation thereof. And "Simulated nanophytoplankton are..." rather than "Nanophytoplankton are..."

**Agree with the suggestion. We will precise better our result's statements.**

**1.7.** P1, Line 27: Suggest replacing "vertical diffusion" with "turbulent vertical diffusion" or "vertical mixing." Diffusion typically refers to molecular diffusion which acts on too small scales to make any difference to the processes considered here.

Agree with the suggestion, change will be done accordingly.

**1.8.** P1, Line 27, sentence starting with "This study highlights the..." This is an overstatement not supported by the results actually presented in this manuscript. See major comment 3.

This statement will be removed.

**1.9.** P2, Line 9: ...because of deleterious impact on coastal ecosystems." The authors should provide one or more references in support of this statement, or modify it. I would like to challenge them to find a study that shows deleterious impacts on the ecosystem in the northern Gulf of Mexico (I am not aware of one). There are studies about specific aspects of the ecosystem, which would be fine to cite if sentence is slightly modified.

Following the reviewer suggestion, we will modify this statement, mentioning the specific

ecosystem aspects that are negatively impacted by bottom hypoxia, such as individual growth and metabolism (Rosas C, et al., 1998; Craig and Crowder, 2005) and stock catchability (Craig, 2012).

**1.10.** P3, Line 5, sentence beginning with "New modelling efforts..." I object to the logic of this statement. Adding complexity to biogeochemical models is not in itself a worthwhile undertaking. It has to be motivated by the scientific questions (e.g. one might be interested in species succession). Sentence should be reformulated accordingly.

*The sentence will be reformulated to:*

New modeling efforts are required to examine spatiotemporal biomass patterns of phytoplankton functional groups across the northern and deep GoM regions.

**1.11.** P3, Line 8: "...diatoms require..." Citing a modeling study (Kishi et al.) in support of a general statement about diatom traits seems inappropriate. There are more appropriate references. I suggest the authors look up publications by Elena Litchman and collaborators. She has worked extensively on documenting phytoplankton functional traits.

We agree with the suggestion. We will include Litchman and Klausmeier (2008) as reference for the diatom traits [Litchman, Elena, and Christopher A. Klausmeier. "Traitbased community ecology of phytoplankton." Annual review of ecology, evolution, and systematics 39 (2008): 615-639.]

**1.12.** P4, Line 30: Which basin does "basin-scale" refer to here?

It refers to the Atlantic Ocean. We will precise the statement in the revised manuscript version.

1.13. P5, Line 18: ...randomly selected year" Which year?

*We will modify the sentence to make the spin-up procedure clearer:*

A 40-year model spin-up was completed before starting the historical simulation. To run the model spin-up, we used the basin-model boundary conditions and ERA surface fluxes of randomly selected years from the 1979-2014 period, following Lee et al. (2011). After the spin-up, the model was run continuously from January 1979 until December of 2014, with monthly averaged fields saved.

**1.14.** P5, Line 23: Stating the model "reproduces" the observations is an overstatement. It would be more appropriate to say they agree qualitatively.

The sentence will be modified to:

*"The spatiotemporal patterns of model and satellite chlorophyll agree qualitatively (Fig. 2)."*

**1.15.** P5, Line 31: The authors should make it much more clear upfront that these are anomalies (i.e. that the bias was removed).

We will indicate that the series in Figure 2 correspond to anomalies with the long-term mean removed.

**2. Response to Referee #2**

**2.1.** My major concern is associated with the validation of the coupled physical biogeochemical model:

First, there is no physical validation presented in the paper, despite that the authors have emphasized the importance of physical processes on the net phytoplankton growth. Has the physical validation work been done and/or published elsewhere? If yes, it is important to summarize that here in some way. If not, I think it's worthwhile to do some extra work on physical validation to make the presented results here more convincing considering how important the physics is controlling the biogeochemical cycling in this region (e.g., the mixing and transport by riverine waters to northern GoM, Loop Current and eddy interactions to deep GoM, etc.). For example, the simulated spatial extent of the high chlorophyll river plume in northern GoM is narrower than that observed in satellite (visually viewed from Fig. 2), could it be associated with the distant transport of riverine nutrients?

We agree that a validation of the physical model component is required. In the revised manuscript version, we will provide model-observations comparisons for SST, coastal sea level anomalies, eddy kinetic energy, salinity, salinity, and T-S vertical profiles in coastal and oceanic domains (see answer to Dr. Fennel, section 1.1.1 and Figures S1-S5).

Second, the validation of biogeochemical (BGC) model doesn't seem sufficient to me. The BGC validation in the paper primarily relies on comparing model simulated and satellite observed surface chlorophyll. While the model overall reproduces the dominant seasonal and spatial patterns in satellite chlorophyll, it significantly underestimates the coastal chlorophyll both in magnitude (2.5-3 times lower in the model) and spatial extent. The authors attribute the mismatch to satellite overestimating in situ observations of chlorophyll in northern GoM. If true, it would be useful to also include comparisons between simulated and in situ observations of chlorophyll in the paper for justification. In addition, while satellite chlorophyll observations have the advantage for model validation due to its spatial and temporal coverage, they are limited to the first optical depth that could hardly represent the plankton dynamics in subsurface water (e.g., the deep chlorophyll maxima). Hence a good complement to the validation might be including comparison to chlorophyll profiles, which to my knowledge is available in GoM during the model simulation period (e.g., the bio-optical profiling float results presented in Green et al., 2014). Also, there are relatively 'abundant' observations, apart from chlorophyll, in the northern GoM, such as those provided by Mechanisms Controlling Hypoxia (MCH) program (http://hypoxia.tamu.edu/field-program), in situ observations of

primary production (Lehrter et al. 2009), and water column community respiration rates (Murrell et al. 2013). These datasets might improve the BGC validation in coastal region where satellite chlorophyll is considered to have higher uncertainty.

We agree that the biogeochemical model requires extra validation and we are working now in the comparison between model and observations for nutrients, plankton biomass, and primary production, using the NODC, GOMRI, and other database available, for both coastal and oceanic domains, which will be included in the revised paper version.

**2.2.** One novelty of this work is that the model includes two phytoplankton types and two zooplankton types that complement the previous modeling work in GoM that mostly only includes one phytoplankton and one zooplankton type. While the additional complexity added to the BGC model is more faithful in representing the lower-trophic level dynamics in real system, it also adds more complexities and challenges in calibrating and validating the model. With respect to calibration, have the parameter values shown in Table 1 (especially those with \*) been informally or formally tuned or optimized? Are the conclusions presented here sensitive to the selected parameter values? I think providing more information/comments on these would be helpful to others. The additional complexity of the BGC model also adds difficulties in model validation, e.g., the modeldata chlorophyll comparison alone cannot tell how reasonable the model simulates each type of phytoplankton group as it could not distinguish the contribution from small- and large-size phytoplankton groups. How has the added complexity benefit us to understand the plankton dynamics in this region? Does the presented model do a better job than the previous modeling work that only include one phytoplankton type (e.g., compared with Xue et al. 2013)? I think readers would appreciate with a bit more discussions/comments on these.

The selected parameters are within ranges reported in previous studies, with \* indicating minor variations from reported values. We agree with the reviewer that more information and comments on the model parameters are helpful to other biogeochemical modelers, and we will include them in the discussion of the new revised version. In addition, we will include a comparison between our biogeochemical model (refer to as GOM model) and Fennel model. This comparison reveals that although Fennel's chlorophyll match better the long-term mean of satellite chlorophyll in the coastal regions, GOM does better reproducing the seasonal chlorophyll patterns in the open ocean, with no significant seasonal bias (see answer to Dr. Fennel, section 1.1.2 and Figures S8-S9). We will run additional experiments to evaluate the sensitivity of the simulated chlorophyll/phytoplankton patterns to changes in model parameters. Specially, we will examine to what degree the improved representation of seasonal chlorophyll patterns in the open ocean region depends on the parameterization of phytoplankton growth (nutrient-limitation and maximum growth rate), zooplankton grazing (micro- and mesozooplankton parameters) and chlorophyll to carbon ratios.

**2.3. Specific comments:**

Page 4, Line 6: Would it be more appropriate to list an observational rather than a modeling work (Xue et al., 2013) as a reference?

We will modify the citation to Green and Gould (2008).

Green, R. E., and R. W. Gould (2008), A predictive model for satellite-derived phytoplankton absorption over the Louisiana shelf hypoxic zone: Effects of nutrients and physical forcing. Journal of Geophysical Research: Oceans, 113(6): 1–17, https://doi.org/10.1029/2007JC004594

Page 4, Line 14: delete one 'to' either in front of the ':' or after the number. *The change will be done accordingly.*

Page 4, Line 22: Why listed MODIS SST here? Has it been used anywhere in the paper? *We apologize for this mistake. However, the revised manuscript version will include a comparison between model SST and MODIS, so a description of the MODIS data will be included in the Data section.*

Page 4, Line 28: Horizontal diffusivity is non-zero here, but it seemed to be neglected when analyzing the role of advection and diffusion in section 3.4.

We did not include horizontal diffusivity term in the budget analysis because is about 2 orders of magnitude smaller than the advection and vertical diffusion terms, so it contribution can be neglected. We will mention this aspect in the revised paper version.

Page 4, Line 30: Does the basin-scale model also include biogeochemistry and provide BGC initial conditions? If not, how do you specify them? Could you also provide more information on how you specify open boundary conditions? Has tide been included? *The basin model specifies the boundary and initial condition for both the physical and biogeochemical model. Tides were not included in the model. The revised manuscript version will have an extended description of all the boundary condition aspects.*

Page 5, Line 18: Where were the boundary conditions and surface fluxes extracted from? the basin-scale model?

The basin model provides the boundary conditions, and the surface fluxes are from ERAinterim (same surface forcing as in the basin model). We will clarify those aspects in the revised version.

Page 6, Line 23: 'mean production values', is it spatial or/and temporal mean? Maybe also provide the standard deviation if available, since the primary production is highly variable?

Standard deviation values of primary production will be provided in the revised manuscript version

Page 7, Line 29: change 'ranges' to 'range'? *The change will be done accordingly.*

Page 8, Line 26: In the text, it's switching between 'summer' (or winter) and 'months' back and forth. Could you specify the summer and winter months at the first time they appear?

We will make the modification accordingly.

Page 10, Line 22-24: This statement is a bit exaggerated to me since the validation is on chlorophyll, a combination of two phytoplankton groups, that how well each type of phytoplankton is simulated by the model is not directly validated.

We agree with the reviewer. We will rewrite this paragraph, indicating the agreement and differences between model and observed chlorophyll patterns. Besides, we will discuss about the comparison between Fennel and our 18-component model.

Fig.2: the lower limit of the color bar is missing? From 0? What does the gray contour line represent? 200m isobath?

The colorbar ranges from 0.05 to 5 mg m-3. We will include the lower limit in the revised Figure 2 version. The contour gray line represents the 200 m isobath. We will mention this in the Figure 2 legend, and also will include '200 m' as contour labels.

Fig.8: should be '...in panels a-b depict...' *Change will be done accordingly*.

**3. Response to Referee #3**

**3.1. Validation of the physical model**

The paper stated that the boundary conditions were from a HYCOM model, yet the model (ROMS)'s own performance regarding circulation and T/S fields was not evaluated, without which, I would have a big question mark about the results presented in the manuscript;

We agree that the physical model component needs to be validated. We have completed part of this validation, which shows a good agreement between observed and simulated patterns of SST, sea level anomalies, eddy kinetic energy, and surface salinity (see section 1.1.1 in answer to Dr. Fennel and Figures S1-S5). In the revised manuscript version, we will also include comparisons of surface salinity time series, as well as vertical profiles of temperature and salinity, in coastal and oceanic domains.

**3.2.** Validation of the biogeochemical model**

The author evaluated their model's performance via a comparison against satellite data and admitted that their model underestimated the Chl-a. And unfortunately, these satellite data were the only source used for model evaluation. How about the model's performance on nutrient and plankton sgroups? Without such information, it is hard to conclude that the model could at least represent the nutrient and biological cycle in the Gulf;

We agree that the biogeochemical model requires extra validation. To evaluate the model performance, we are working now in comparing pattern in nutrients, plankton biomass,

primary production, and chlorophyll with observations from NODC, GOMRI, and other database available. This validation will be included in the revised paper version.

**3.3.** Given that point 1) and point 2) were addressed, I could not find the benefit of introducing the new plankton group (2 phytoplankton and 3 zooplankton vs. 1 phytoplankton and 1 zooplankton by Fennel at al. 2011), which, indeed, could be the most important contribution of this study

We understand the reviewer concern regarding to the benefit of introducing a more complex representation of lower trophic levels, as the first manuscript version did not include any contrast with outputs from simpler biogeochemical models. In the revised manuscript version, we will include a comparison between chlorophyll patterns derived from our 18-component model (refers to as GOM) and Fennel's model, which reveals that GOM does better reproducing the seasonal chlorophyll patterns in the open ocean region, with no significant seasonal bias (see section 1.1.2 in answer to Dr. Fennel and Figures S8-S9). This result strongly suggests that our model allows a better representation of phytoplankton dynamics compared to previous model results in the open ocean region. To complement this comparison, we will perform new sensitivity experiments, examining to what degree the improved representation of seasonal chlorophyll patterns in the open ocean region depends on the parameterization of phytoplankton growth (nutrient-limitation and maximum growth rate), zooplankton grazing (micro- and mesozooplankton parameters) and chlorophyll to carbon ratios.

**References**

Behrenfeld, M. (2010) Abandoning Sverdrup's critical depth hypothesis on phytoplankton blooms, Ecology 91(4), 977-989.

Bode, A. and Dortch, Q. (1996) Uptake and regeneration of inorganic nitrogen in coastal waters influenced by the Mississippi River: spatial and seasonal variations. J. Plankton Res. 18, 2251–2268.

Craig, J.K. (2012) Aggregation on the edge: Effects of hypoxia avoidance on the spatial distribution of brown shrimp and demersal fishes in the northern Gulf of Mexico. Mar Ecol Prog Ser 445:75–95

Craig J.K, Crowder L.B (2005) Hypoxia-induced habitat shifts and energetic consequences in Atlantic croaker and brown shrimp on the Gulf of Mexico shelf. Mar Ecol Prog Ser 294:79–94.

Dagg, M.J., and Breed, G.A. (2003) Biological effects of Mississippi River nitrogen on the northern Gulf of Mexico—a review and synthesis, Journal of Marine Systems, 43(3), 133-152.

Fennel, K., Hetland, R., Feng, Y., and DiMarco, S. (2011) A coupled physical-biological model of the Northern Gulf of Mexico shelf: model description, validation and analysis of phytoplankton variability, Biogeosciences, 8(7), 1881.

Fennel, K. and Laurent, A. (2017) N and P as ultimate and proximate limiting nutrients in the northern Gulf of Mexico: Implications for hypoxia reduction strategies, Biogeosciences Discuss., https://doi.org/10.5194/bg-2017-470, in review.

Laurent, A., Fennel, K., Hu, J., and Hetland, R. (2012) Simulating the effects of phosphorus limitation in the Mississippi and Atchafalaya River plumes, Biogeosciences, 9(11), 4707-4723.

Laurent, A., and Fennel, K. (2014). Simulated reduction of hypoxia in the northern Gulf of Mexico due to phosphorus limitation. Elem Sci Anth, 2.

Litchman, E., and Klausmeier, C.A. (2008) Trait-based community ecology of phytoplankton, Annual review of ecology, evolution, and systematics 39: 615-639.

Marta-Almeida, M., Hetland R.D., and Zhang, X. (2013) Evaluation of model nesting performance on the Texas-Louisiana continental shelf. J.Geophysical Res.-Oceans, 118: 1-16. doi: 10.1002/jgrc.20163.

Nababan, B. (2005) Bio-optical variability of surface waters in the Northeastern Gulf of Mexico. PhD thesis, College of Marine Science, University of South Florida, Tampa, FL, 145 pp.

Nelson D.M., and Dortch, Q. (1996) Silic acid and silicon limitation in the plume of the Mississippi River: evidence from kinetic studies in spring and summer, Marine Ecology Progress Series, 136, 163-178.

Rosas C, et al. (1998) Effect of dissolved oxygen on the energy balance and survival of Penaeus setiferus juveniles. Mar Ecol Prog Ser 174:67–75.

Sylvan, J.B., Dortch, Q., Nelson, D.M., Brown, A.F.M., Morrison, W., and Ammerman, J.W. (2006) Phosphorus limits phytoplankton growth on the Louisiana shelf during the period of hypoxia formation, Environ. Sci. Technol., 40, 7548–7553, doi:10.1021/es061417t.

Sylvan, J. B., Quigg, A., Tozzi, S., and Ammerman, J. W. (2007) Eutrophication-induced phosphorus limitation in the Mississippi River Plume: evidence from fast repetition rate fluorometry, Limnol. Oceanogr., 52, 2679–2685, doi:10.4319/lo.2007.52.6.2679.

Xue, Z., He, R., Fennel, K., Cai, W. J., Lohrenz, S., and Hopkinson, C. (2013) Modeling ocean circulation and biogeochemical variability in the Gulf of Mexico, Biogeosciences, 10(11), 7219.

Zhao, Y., and Guigg A. (2014) Nutrient limitation in Northern Gulf of Mexico (NGOM): phytoplankton communities and photosynthesis respond to nutrient pulse, PloS one 9.2, e88732.

Figure. S1. Monthly time series of SST derived from model outputs and MODIS for the Mississippi delta, Texas shelf, and Deep Gulf regions. Correlation coefficient between model and MODIS series are indicated at each panel.

---

## Author Response (AR1)

**Letter to the Editor**

Dear Dr. Emmanuel Boss,

*We highly appreciate the opportunity for submitting a revised version of our manuscript entitled "Seasonal Patterns in Phytoplankton Biomass across the Northern and Deep Gulf of Mexico: A Numerical Model Study", co-authored by Sang-Ki Lee, Yanyun Liu, Frank J. Hernandez Jr., Frank E. Muller-Karger, and John T. Lamkin. We are thankful for all the valuable reviewer comments and suggestions. Please find attached the new version of the responses to the reviewers, as well as a new manuscript version with the 'track change' option. In the revised manuscript version, we have addressed all the reviewers' indications, including a validation of the physical and biogeochemical model components. During the validation process, we detected an error in the prescribed boundary conditions for silicate, which produced a significant underestimation of the silicate concentration in the deep Gulf of Mexico (GoM). Consequently, we had to re-run the model for the entire study period correcting for the right silicate boundary conditions. For this new model run, we did minor changes in the model parameters, which help to reduce part of the disagreement between the simulated and satellite chlorophyll in the coastal regions. The results from the new model run do not modify the main finding described in the previous paper version, but the silicate limitation patterns. We obtained that model diatom growth in the deep GoM is silicate-limited only during winter, and not year-round as in the previous version.*

*We believe that the manuscript is significantly improved, and we hope it is suitable for publication in Biogeosciences.*

*Sincerely,*

*Fabian Gomez, on behalf of the co-authors.*

**1. Response to Dr. Fennel**

**1.1 Validation**

1.1.1. Physical validation:

The authors provide no validation of the physical model. If there are previous publications in which this is reported, it would be fine to refer to those. Otherwise some physical model validation should be provided.

*We agree that the manuscript needs some physical-component validation. In this new manuscript version we have included a validation of the physical model in the Appendix, which includes model-data comparison of SST, coastal sea level anomalies, eddy kinetic energy, surface salinity, and vertical profiles of temperature, salinity, and density.*

1.1.2. Chlorophyll patterns:

On page 6 (first paragraph) the authors state that the model underestimates mean satellite chlorophyll by factors between 2.5 to 3. These are rather large deviations in the mean. They then give reasons for why the satellite can't be trusted. There are two problems with this: first, the simpler models of Fennel et al. (2011) and Laurent et al. (2012) reproduced satellite chlorophyll without such large biases; and, second, if chlorophyll cannot be trusted it shouldn't be used in validation. However, satellite derived chlorophyll is essentially the only data set that is used in this manuscript to validate the model.

*We understand the concern pointed out by the reviewer regarding to the chlorophyll validation section. We have included now a comparison between model outputs and in situ observations derived from the Coastal Waters Consortium (Rabalais, 2015; Smith, 2015), as well as from APEX profiling floats (Hamilton and Leidos, 2017). The comparison between in-situ and satellite chlorophyll in the Louisiana-Texas shelf actually suggests chlorophyll overestimation by the satellite sensors in the MS delta region during fall-winter.*

*Regarding to the first related problem (a simpler model reproduces better the chlorophyll variability), we agree that the Fennel model matches better the chlorophyll pattern in the Louisiana-Texas shelf, as shown by Fennel et al. (2011) and Laurent et al. (2012). However, the Fennel model tends to overestimates satellite chlorophyll in the open ocean region, which can be noted in the chlorophyll patterns reported by Xue et al. (2013). We have included a direct comparison between our biogeochemical model (hereinafter GoMBio) and Fennel model, which is presented in the Appendix version.*

*Regarding to the second problem (if chlorophyll cannot be trusted it shouldn't be used in validation), previous studies have shown that although satellite chlorophyll can largely overestimate in situ chlorophyll in regions influenced by river runoff (until by 300%), there is a significant correlation between satellite and in situ chlorophyll (as example, see Figure1.10 in Nababan, 2005). Besides, the bias associated to optically complex waters occurs mainly in the coastal region but not in Open Ocean. These two aspects, added to the fact that satellite sensors provide large spatial and temporal coverage to properly describe seasonal and interannual variability, makes the model-satellite comparison of chlorophyll valid.*

1.1.3. Biomass and nutrients:

There should be some validation of the biogeochemical model with in situ observations of phytoplankton biomass and nutrients.

Such observations are available in the NODC and GOMRI databases. Profile comparisons for the open Gulf should be included.

*In the new manuscript version we have included a comparison with satellite chlorophyll, in-situ chlorophyll derived from the Coastal Waters Consortium (Rabalais, 2015; Smith, 2015) and APEX profiling floats (Hamilton and Leidos, 2017), diatom to total chlorophyll ratios reported by Zhao and Quigg (2014), primary production (Lehrter et al., 2009; Biggs, 1992, Sanchez, 1992), and nutrients (Rabalais, 2015; Smith, 2015; Parson et al, 2015; Wanninkhof et al., 2014).*

1.1.4. Diatom and nanophytoplankton:

Perhaps the largest omission, given the objective of the study, is that there is no validation of the different phytoplankton groups. I recognize that it is hard to get good data sets for this purpose, but there are some algorithms that can be used to separate satellite-derived chlorophyll into different size groups (see Hirata et al. 2011, Mouw et al. 2017 and references therein).

*As Dr. Fennel indicates, it is difficult to obtain data set to validate the diatom and nanophytoplankton components. We are not including a satellite-based analysis on phytoplankton group biomass pattern, as time limitations preclude performing this type of analysis (which we think can be the topic for an independent paper). However, we have used chlorophyll data of functional phytoplankton group reported by Zhao and Quigg (2014). Based on these data, we estimate the diatom to total chlorophyll ratio for two stations in the Louisiana shelf. The model-data comparison showed that our biogeochemical model does reasonable well reproducing the observed diatom ratios, including a documented diatom chlorophyll decline during summer (see paper Fig. 5).*

**1.2. Terminology**

With regard to terminology: In section 3.4 the authors define the "biological term" as the balance between phytoplankton production and biological losses. This is the same as Net Phytoplankton Growth, a widely used term in biological oceanography. It is not only unnecessary to redefine this as a new term, but also potentially confusing. Then the authors state that the balance of the biological and physical terms determines the change in net phytoplankton growth. This is wrong. Net phytoplankton growth is equal to what they defined as the biological term. The balance between the biological and physical terms is the local rate of change of phytoplankton.

*We highly appreciate the term names clarification, and understand that is preferable avoid any confusing terminology in the budget analysis. We modified the term names following Dr. Fennel indications.*

**1.3. Improving description of the results and the context in the existing literature**:

1.3.1. In the last sentence of the abstract, they claim that their study shows the importance of representing large and small plankton in order to describe PP patterns. This is not supported by the results presented. On the one hand, there is no validation of the contributions of large and small phytoplankton to biomass and PP (see 1 above). On the other hand, there is no comparison to simulated phytoplankton abundance and PP from a model with only one phytoplankton group. Simpler models exist that, in fact, reproduce chlorophyll from satellite more accurately than this model in the Mississippi plume region (see comment 1.2).

*We agree that that sentence is not supported by the result presented and was removed from the abstract.*

1.3.2. In the second to fourth sentences of the discussion the authors make statements about their results that are not supported. "Inclusion of two phytoplankton components allowed for realistic representation..." is not accurate as simple models arguably reproduced this better (see comment 1.2). "The good agreement between model outputs and observations of chlorophyll..." is a questionable statement (see again comment 1.2).

*We decided to tone down our statement and reformulate the entire first discussion by:*

*"We configured an ocean-biogeochemical model for the GoM that explicitly represents two types of phytoplankton and zooplankton, and nitrogen and silica as limiting nutrients for phytoplankton growth. Our model reproduces reasonable well the main physical and biochemical patterns, although an underestimation of the mean surface chlorophyll is evident in the northern shelf, especially on bottom depth < 20 m. A comparison with in situ chlorophyll observations suggests that part of the model-satellite chlorophyll disagreement could be linked to chlorophyll overestimation by satellite sensors during fall-winter. Realistic representation of phytoplankton variability in region with strong physical and biochemical gradients, like those in the northern GoM, is challenging. Previous modeling efforts on the Louisiana-Texas shelf based on Fennel's model reproduced better the mean satellite chlorophyll condition than our model (e.g. Fennel et al., 2011; Laurent et al., 2012). However, Fennel's model tends to overestimate satellite chlorophyll by a factor of >3 in the Deep Ocean region during winter, which could be linked to misrepresentation of microzooplankton grazing (see section 4 in Appendix). We acknowledge that additional components and processes could be included in our model, such as phosphorus cycling, iron limitation and nitrogen fixation, to represent more realistic biogeochemical dynamics. We also recognize that more observational studies will be required to constrain better our model parameters, as well as the biogeochemical fluxes between land and ocean. Nevertheless, we believe that the current model configuration can capture well enough the seasonal dynamics of diatoms and nanophytoplankton biomass in the GoM. It is known that variations in phytoplankton composition can have important repercussion for the ecosystem, including changes in upper trophic levels dynamics, carbon export (carbon export is enhanced in diatom-dominated food webs) and bottom hypoxia (Dagg et al., 2003; Green et al., 2008). Therefore, modeling efforts exploring variability in phytoplankton component, such as this study, are needed to advance our understanding of ecosystem variability in the GoM."*

1.3.3. With respect to phosphorus (P) the authors seem to be diminishing previous findings in an effort to justify why their model does not include P. On page 3 (line 8, sentence starting with "Although: : :") they seem to suggest that previous studies (specifically Laurent et al. 2012) suggest P limitation to be unimportant. This is not the conclusion of Laurent et al. (2012) nor of the follow-up studies by Laurent and Fennel (2014) and Fennel and Laurent (2017), which are consistent with the observational studies by Sylvan et al. (2006, 2007). All these studies do suggest the P limitation is critically important in the region influenced by the Mississippi River plume. Saying that P limitation is "moderate" while N and Si limitation are "critical" seems disingenuous. To be clear, I do not object to the fact that P is neglected in this model. All models are simplifications. It would be fine to state that their model neglects P, although it has been shown to be important in a portion of the model region. In the Discussion (end of first paragraph) it would be appropriate to be more forthcoming about previous studies on P limitation.

*We understand the point indicated by the reviewer and recognize that phosphorous limitation can be critical in the region influenced by the Mississippi River. We modified the statement in the 'Model description' section to:*

*"The model does not include phosphate as limiting nutrient for phytoplankton growth. Although previous studies have indicated the existence of phosphate limitation near the MS-A deltas during May-July (Sylvan et al. 2006, 2007; Laurent et al., 2012; Laurent and Fennel, 2014; Fennel and Laurent, 2017), we focus here on the role of N and Si, as observational studies suggest that N and Si can modulate phytoplankton production and composition across the northern GoM (Dortch and Whitledge, 1992; Nelson and Dortch, 1996; Lohrenz et al., 1997; 2008; Rabalais et al., 2002; Zhao and Quigg, 2014)."*

1.3.4. The statement in the Discussion (last sentence starting on page 11) about consistency with the dilution-recoupling hypothesis of Behrenfeld seems a bit cavalier. No detailed analysis in support of this statement was presented in this manuscript. The authors may want to consider the study by Kuhn et al. (2015), which used the same data set as Behrenfeld, and later papers by Behrenfeld where he backtracked himself somewhat from his early paper (Behrenfeld et al. 2013).

*We agree with the reviewer that more analyses need to be done to support the consistency between the dilution-recoupling hypothesis and our model results. Since this aspect is beyond the paper goals, we decided to remove the dilution-recoupling hypothesis part from the discussion.*

**Other comments (not in order of importance)**:

**1.4.** P1, Line 13: Suggest inserting "improving" after "tools for"

*The change was done accordingly.*

**1.5.** P1, Line 14: Suggest removing "However"

*The change was done accordingly.*

**1.6.** P1, Line 19, sentence starting with "The model results show..." and following sentences in the abstract. Because diatoms in the model are strongly silica-limited doesn't necessarily mean they are in reality. Making inferences about reality from the model requires that the model accurately reproduces reality, which in this case is hard to prove. The authors certainly haven't (see my comments about validation). I would suggest that here and throughout the remainder of the abstract and manuscript the authors are more precise in their language. It is fine to say "diatoms in the model are silica limited" or some variation thereof. And "Simulated nanophytoplankton are..." rather than "Nanophytoplankton are..."

*Agree with the suggestion. We precise better our result's statements in the revised paper version.*

**1.7.** P1, Line 27: Suggest replacing "vertical diffusion" with "turbulent vertical diffusion" or "vertical mixing." Diffusion typically refers to molecular diffusion which acts on too small scales to make any difference to the processes considered here.

*Agree with the suggestion, change was done accordingly.*

**1.8.** P1, Line 27, sentence starting with "This study highlights the..." This is an overstatement not supported by the results

actually presented in this manuscript. See major comment 3.

*This statement was removed.*

**1.9.** P2, Line 9: ...because of deleterious impact on coastal ecosystems." The authors should provide one or more references in support of this statement, or modify it. I would like to challenge them to find a study that shows deleterious impacts on the ecosystem in the northern Gulf of Mexico (I am not aware of one). There are studies about specific aspects of the ecosystem, which would be fine to cite if sentence is slightly modified.

*Following the reviewer suggestion, we modified this statement, mentioning the specific ecosystem aspects that are negatively impacted by bottom hypoxia, such as individual growth and metabolism (Rosas C, et al., 1998; Craig and Crowder, 2005) and specie distribution (Craig, 2012).*

**1.10.** P3, Line 5, sentence beginning with "New modelling efforts..." I object to the logic of this statement. Adding complexity to biogeochemical models is not in itself a worthwhile undertaking. It has to be motivated by the scientific questions (e.g. one might be interested in species succession). Sentence should be reformulated accordingly.

*The sentence was reformulated to:*
*"New modeling efforts are required to examine spatiotemporal patterns of main phytoplankton functional groups across the northern and deep GoM."*

**1.11.** P3, Line 8: "...diatoms require..." Citing a modeling study (Kishi et al.) in support of a general statement about diatom traits seems inappropriate. There are more appropriate references. I suggest the authors look up publications by Elena Litchman and collaborators. She has worked extensively on documenting phytoplankton functional traits.

*We agree with the suggestion. We included Litchman and Klausmeier (2008) as reference for the diatom traits [Litchman, Elena, and Christopher A. Klausmeier. "Trait-based community ecology of phytoplankton." Annual review of ecology, evolution, and systematics 39 (2008): 615-639.]*

**1.12.** P4, Line 30: Which basin does "basin-scale" refer to here?

*It refers to the Atlantic Ocean. We precise the statement in the revised manuscript version.*

**1.13.** P5, Line 18: ...randomly selected year" Which year?

*We modified the sentence to make the spin-up procedure clearer:*

*"A 40-year model spin-up was completed before starting the historical simulation. To spin-up the model, we used the basin-model boundary conditions and the ERA surface fluxes of randomly selected years from 1979-2014, following Lee et al. (2011)."*

**1.14.** P5, Line 23: Stating the model "reproduces" the observations is an overstatement. It would be more appropriate to say they agree qualitatively.

*The sentence was modified to:*

*"Modeled surface chlorophyll agreed qualitatively well in the spatiotemporal patterns with the satellite chlorophyll (Fig. 2)."*

**1.15.** P5, Line 31: The authors should make it much more clear upfront that these are anomalies (i.e. that the bias was removed).

*We decided presenting the chlorophyll time series without removing the long-term mean, and make easier the comparison with observations (Fig. 3).*

**1.16.** Results, general: No oxygen results are shown. Given this, there is not much point saying the model includes oxygen.

*We agree with Dr. Fennel so we excluded oxygen from the model.*

**2. Response to Referee #2**

**2.1.** My major concern is associated with the validation of the coupled physical biogeochemical model:

First, there is no physical validation presented in the paper, despite that the authors have emphasized the importance of physical processes on the net phytoplankton growth. Has the physical validation work been done and/or published elsewhere? If yes, it is important to summarize that here in some way. If not, I think it's worthwhile to do some extra work on physical validation to make the presented results here more convincing considering how important the physics is controlling the biogeochemical cycling in this region (e.g., the mixing and transport by riverine waters to northern GoM, Loop Current and eddy interactions to deep GoM, etc.). For example, the simulated spatial extent of the high chlorophyll river plume in northern GoM is narrower than that observed in satellite (visually viewed from Fig. 2), could it be associated with the distant transport of riverine nutrients?

*We agree that a validation of the physical model component is required. In the revised manuscript version, we have included in the Appendix a validation for SST, coastal sea level anomalies, eddy kinetic energy, surface salinity, and vertical profiles of temperature, salinity, and density.*

Second, the validation of biogeochemical (BGC) model doesn't seem sufficient to me. The BGC validation in the paper primarily relies on comparing model simulated and satellite observed surface chlorophyll. While the model overall reproduces the dominant seasonal and spatial patterns in satellite chlorophyll, it significantly underestimates the coastal chlorophyll both in magnitude (2.5-3 times lower in the model) and spatial extent. The authors attribute the mismatch to satellite overestimating in situ observations of chlorophyll in northern GoM. If true, it would be useful to also include comparisons between simulated and in situ observations of chlorophyll in the paper for justification. In addition, while satellite chlorophyll observations have the advantage for model validation due to its spatial and temporal coverage, they are limited to the first optical depth that could hardly represent the plankton dynamics in subsurface water (e.g., the deep chlorophyll maxima). Hence a good complement to the validation might be including comparison to chlorophyll profiles, which to my knowledge is available in GoM during the

model simulation period (e.g., the bio-optical profiling float results presented in Green et al., 2014). Also, there are relatively 'abundant' observations, apart from chlorophyll, in the northern GoM, such as those provided by Mechanisms Controlling Hypoxia (MCH) program (http://hypoxia.tamu.edu/field-program), in situ observations of primary production (Lehrter et al. 2009), and water column community respiration rates (Murrell et al. 2013). These datasets might improve the BGC validation in coastal region where satellite chlorophyll is considered to have higher uncertainty.

*We have included now a validation section for the biogeochemical model. We performed model-data comparison of chlorophyll, primary production, diatom to total chlorophyll ratios, and nutrients.*

**2.2.** One novelty of this work is that the model includes two phytoplankton types and two zooplankton types that complement the previous modeling work in GoM that mostly only includes one phytoplankton and one zooplankton type. While the additional complexity added to the BGC model is more faithful in representing the lower-trophic level dynamics in real system, it also adds more complexities and challenges in calibrating and validating the model. With respect to calibration, have the parameter values shown in Table 1 (especially those with **\***) been informally or formally tuned or optimized? Are the conclusions presented here sensitive to the selected parameter values? I think providing more information/comments on these would be helpful to others. The additional complexity of the BGC model also adds difficulties in model validation, e.g., the model-data chlorophyll comparison alone cannot tell how reasonable the model simulates each type of phytoplankton group as it could not distinguish the contribution from small- and large-size phytoplankton groups. How has the added complexity benefit us to understand the plankton dynamics in this region? Does the presented model do a better job than the previous modeling work that only include one phytoplankton type (e.g., compared with Xue et al. 2013)? I think readers would appreciate with a bit more discussions/comments on these.

*The selected parameters are within ranges reported in previous studies, with \* indicating minor variations from reported values. We agree with the reviewer that more information and comments sensitivity analysis and comparison with previous model can be helpful to other biogeochemical modelers. In the Appendix section we have included a direct comparison to Fennel's model, where we pointed out that although Fennel model is able to catch better the mean satellite chlorophyll condition in the coastal region, it overestimates satellite chlorophyll during winter in the Deep Ocean region. A coupled of sensitivity analyses were performed to show that this winter overestimation could be linked to misrepresentation of zooplankton grazing in the Deep Ocean region.*

**2.3. Specific comments:**

Page 4, Line 6: Would it be more appropriate to list an observational rather than a modeling work (Xue et al., 2013) as a reference?
*We modified the citation to Green and Gould (2008).*
*Green, R. E., and R. W. Gould (2008), A predictive model for satellite-derived phytoplankton absorption over the Louisiana shelf hypoxic zone: Effects of nutrients and physical forcing. Journal of Geophysical Research: Oceans, 113(6): 1–17, https://doi.org/10.1029/2007JC004594*

Page 4, Line 14: delete one 'to' either in front of the ':' or after the number.

*The change was done accordingly.*

Page 4, Line 22: Why listed MODIS SST here? Has it been used anywhere in the paper?

*We apologize for this mistake. We have corrected it in the new manuscript version.*

Page 4, Line 28: Horizontal diffusivity is non-zero here, but it seemed to be neglected when analyzing the role of advection and diffusion in section 3.4.

*We did not include horizontal diffusivity term in the budget analysis because is 2 orders of magnitude smaller than the advection and vertical diffusion terms, so it contribution can be neglected. We mentioned this aspect in new Figure's legend of the Budget Analysis (current Fig. 12).*

Page 4, Line 30: Does the basin-scale model also include biogeochemistry and provide BGC initial conditions? If not, how do you specify them? Could you also provide more information on how you specify open boundary conditions? Has tide been included?

*The basin model specifies the boundary and initial condition for both the physical and biogeochemical model. Tides were not included in the model. We have precise better those aspects in the revised Data and Model section.*

Page 5, Line 18: Where were the boundary conditions and surface fluxes extracted from? the basin-scale model?

*The basin model provides the boundary conditions, and the surface fluxes are from ERA-interim (same surface forcing as in the basin model). We clarified those aspects in the revised version.*

Page 6, Line 23: 'mean production values', is it spatial or/and temporal mean? Maybe also provide the standard deviation if available, since the primary production is highly variable?

*To validate our simulated production rates, we show in current Figure 6 boxplots for the simulated and observed primary productivity rates in the MS delta, Texas shelf, and Deep Ocean regions.*

Page 7, Line 29: change 'ranges' to 'range'?

*The change was done accordingly.*

Page 8, Line 26: In the text, it's switching between 'summer' (or winter) and 'months' back and forth. Could you specify the summer and winter months at the first time they appear?

*We now opt for using month intervals instead of seasons.*

Page 10, Line 22-24: This statement is a bit exaggerated to me since the validation is on chlorophyll, a combination of two phytoplankton groups, that how well each type of phytoplankton is simulated by the model is not directly validated.

*We agree with the reviewer. We re-wrote completely the first paragraph from the Discussion section, making more accurate our statements.*

Fig.2: the lower limit of the color bar is missing? From 0? What does the gray contour line represent? 200m isobath?

*The lower and upper limits for the color bar are now indicated. The contour grey line represents the 200 m isobath. We mentioned this in the Figure 2 legend.*

Fig.8: should be '...in panels a-b depict...'
*Change was done accordingly.*

==3. Response to Referee #3==

**3.1. Validation of the physical model**

The paper stated that the boundary conditions were from a HYCOM model, yet the model (ROMS)'s own performance regarding circulation and T/S fields was not evaluated, without which, I would have a big question mark about the results presented in the manuscript;

*We included now a validation of the physical model in the Appendix section. That includes data-model comparisons for SST, sea level anomalies, eddy kinetic energy, surface salinity, and vertical profiles of temperature, salinity, and density.*

**3.2. Validation of the biogeochemical model**

The author evaluated their model's performance via a comparison against satellite data and admitted that their model underestimated the Chl-a. And unfortunately, these satellite data were the only source used for model evaluation. How about the model's performance on nutrient and plankton sgroups? Without such information, it is hard to conclude that the model could at least represent the nutrient and biological cycle in the Gulf;

*We have included model-data comparisons in the revised paper version. We used satellite chlorophyll data, in-situ chlorophyll from the Coastal Waters Consortium (Rabalais, 2015; Smith, 2015) and APEX profiling floats (Hamilton and Leidos, 2017), diatom to total chlorophyll ratios reported by Zhao and Quigg (2014), primary production (Lehrter et al., 2009; Biggs, 1992, Sanchez, 1992), and nutrients (Rabalais, 2015; Smith, 2015; Parson et al, 2015; Wanninkhof et al., 2014).*

**3.3.** Given that point 1) and point 2) were addressed, I could not find the benefit of introducing the new plankton group (2 phytoplankton and 3 zooplankton vs. 1 phytoplankton and 1 zooplankton by Fennel at al. 2011), which, indeed, could be the most important contribution of this study

*We understand the reviewer concern regarding to the benefit of introducing a more complex representation of lower trophic levels, as the first manuscript version did not include any contrast with outputs from simpler biogeochemical models. In the revised manuscript version, we have included a comparison between chlorophyll patterns derived from our model (GoMBio), Fennel's model, and SeaWiFS, which shows that our model simulates better the seasonal patterns of chlorophyll in the deep GoM (see Appendix, section 4). Sensitivity analysis suggests that representation of microzooplankton grazing is relevant to modulate the amplitude of the winter chlorophyll peak in that region.*

*Besides that comparison, we consider that biogeochemical modeling efforts addressing more complex lower-trophic level dynamics are needed in the GoM, as observational studies suggests that changes in plankton composition can significantly*

*impact on upper-trophic levels abundance/distribution, carbon export, remineralization process, and bottom hypoxia (e.g. Dagg et al., 2003; Green et al., 2008). We recognize that still we need more observational studies to constrain better model parameters, and that additional model component and processes may be required to improve the representation of biogeochemical dynamics. However, we believe that our study gives an initial framework, which is a step forward to advance our understanding of phytoplankton functional group dynamics in the region. This aspect of our modeling effort is pointed out in the revised Discussion section (see first paragraph in Discussion section).*

***Additional author's comments***

*Figures 5, 8, and 12 from previous paper version were eliminated, as four new figures were added in the new validation section. Few paragraphs from the previous version that we consider less relevant for the current paper narrative were also discarded.*

*The extension of the three sub-regions used to describe the model phytoplankton patterns were slightly changed from previous version. The reason of this change was trying to encompass zones with in-situ observations, in order to make more robust the model-data comparison. The MS delta region was extended west from 90.8°W to 91.4°W, the east limit of the Texas shelf region was extended east from 92.25°W to 92°W, and the Deep Ocean region was extended east from 86.5°W to 85.5W.*

***References***

[revised manuscript text omitted]

---

## Author Response (AR2)

**Letter to the Editor**

Dear Dr. Emmanuel Boss,

Thanks for the time and effort spent in the revision process of our manuscript. We highly appreciate all the reviewers comments and suggestions, which led to a significantly improvement of the paper. In this new manuscript version we modified the biomass fluxes terminology, using net phytoplankton growth instead of the 'biological term', and advection plus mixing instead of 'physical term'. We also corrected the minor spelling mistakes. Please find attached the new manuscript version with the last changes tracked.

Sincerely,

Fabian Gomez, on behalf of the co-authors.

[revised manuscript text omitted]
 net phytoplankton growth, which is the balance between production and biological losses ( Fig. 11d-f). The net phytoplankton growth  displays distinct patterns for each phytoplankton component and region. The maximum net growth  for diatom is in January-February in the MS delta, December-January in the Texas shelf, and February in the Deep Ocean, while the maximum net growth

10  for nanophytoplankton is in April in the MS delta, February in the Texas shelf, and January in the Deep Ocean. The net growth  for diatoms and nanophytoplankton begins to decline before the production maximum. Moreover, in the Texas shelf, the net growth  is negative during the production maximum. In the three regions, the net growth  for total phytoplankton (diatoms plus nanophytoplankton) is positive in November-February, has a marked decline in spring, and is negative in May-August. The seasonality of the net phytoplankton growth

15 contrasts with the pattern in the SGR in the MS delta and Texas shelf, as SGR is minimum in December-January and maximum in June. All these features suggest that the seasonal changes in model phytoplankton biomass are strongly modulated by biological losses. Zooplankton grazing is the dominant biological loss term (Fig. 11g-i), markedly prevailing upon mortality and exudation (not shown). Microzooplankton exert the strongest grazing pressure on nanophytoplankton biomass, and mesozooplankton on diatoms, with the grazing patterns closely following the patterns in production. The

20 seasonal patterns for microzooplankton (mesozooplankton) grazing upon nanophytoplankton (diatoms) closely follow the patterns in nanophytoplankton (diatom) production. Peaks in micro- and mesozooplankton grazing are concomitant or lag by 1 month the peak in nanophytoplankton and diatom production.

     The seasonal patterns in  net phytoplankton growth  do not completely explain the seasonal changes in model phytoplankton biomass. To fully elucidate the local phytoplankton biomass change, the role of physically driven

25 fluxes of phytoplankton biomass  needs to be examined. To this effect, we estimate the  advection + mixing term, which  represents the sum of advection and turbulent diffusion of phytoplankton biomass, and compare it with the net phytoplankton growth  (
[revised manuscript text omitted]

**Data Availability**

The ocean-biogeochemical model outputs used in this study are available in the Network Common Data Form (NetCDF) format at the NOAA-AOML server.

**Conflicts of interests**

The authors declare that they have no conflict of interest.

**Acknowledgments**

We would like to thank Chris Kelble for his thoughtful comments and suggestions. We also thank Dr. Katja Fennel and the two anonymous reviewers for their useful comments, which led to a significant improvement of the paper. This work was supported by the Northern Gulf Institute (NGI grants: 15-NGI2-119 and 16-NGI3-14), the base funding of NOAA AOML, and the NOAA Ocean Acidification Program.

[revised manuscript text omitted]